# Proteomic and Metabolomic Profiling Elucidate the Impact of PEDV on Yorkshire Piglets and Reveal the Underlying Molecular Mechanism of PEDV Response

**Lijun Shi [1], Huihui Li [1], Chunxiang Zhou [2] and Lixian Wang [1,\*]**

[1]  Institute of Animal Science, Chinese Academy of Agricultural Sciences, Beijing 100193, China; shilijun01@caas.cn (L.S.); 82101202342@caas.cn (H.L.)
[2]  Medical College, Huanghe Science and Technology University, Zhengzhou 450061, China; zcx8008@hhstu.edu.cn
\*  Correspondence: iaswlx@263.net

**Abstract:** Porcine epidemic diarrhea virus (PEDV) is an RNA virus causing acute diarrhea, vomiting, dehydration and high mortality in piglets, and poses significant challenges for the global pig industry. However, the molecular mechanism underlying PEDV in piglets has not been well comprehended. In this study, we constructed the Yorkshire piglet model (control, resistance, and susceptible groups) to perform the proteomics of jejunum and metabolomics of serum. In total, 734 differentially expressed proteins (DEPs) and 208 significant differentially accumulated metabolites (DAMs) were identified, and the functional annotation showed that they were mainly involved in metabolic and signal transduction pathways. Further, we used weighted gene co-expression network analysis (WGCNA), short timeseries expression miner (STEM) and random forest analyses to detect key promising biomarkers for each corresponding group. By constructing the relationships between proteins and metabolites, we excavated the biomarkers and explained their molecular mechanism of the PEDV response. These data and results might be good resources for the PEDV infection and offer valuable insights into the molecular PEDV response mechanisms.

**Keywords:** PEDV; jejunum; Yorkshire piglets; proteomic; metabolomic





## 1. Introduction

Diarrhea has been a persistent global problem in piglet farming for a considerable period. Various factors contribute to piglet diarrhea, such as nutrition and infectious agents. Notably, porcine epidemic diarrhea virus (PEDV) is a primary culprit responsible for intestinal damage in piglets [1]. Although the fecal–oral route is commonly acknowledged, an escalating number of reports suggest that airborne transmission may play a role in the spread of PEDV outbreaks [2]. PEDV, as an RNA virus classified in the genus Alphacoronavirus, family coronaviridae [3], causes acute diarrhea, vomiting, dehydration and high mortality in piglets. PEDV, widely transmitted over an extended period, has undergone mutations that enhance its virulence, infectivity, and pathogenicity. This poses significant challenges for the global pig industry, as these evolved variations of PEDV present increased strength in causing diseases and have potent detrimental effects [4]. Due to the ongoing mutation of the PEDV genome and the limited ability to induce mucosal immunity, an effective vaccine against PEDV is still lacking [5]. Investigation into the receptors facilitating the entry of PEDV into mammalian cells has proven critical [4], and strides have been made in the identification of receptor proteins for PEDV, such as pAPN, which might act as a receptor facilitating virus entry and determining tissue tropism [6]. However, the definitive receptor protein for PEDV has not been conclusively determined, and the mechanisms behind PEDV pathogenesis and immunomodulation remain largely undisclosed.

The essential flow of information in biological systems goes from DNA (genome) to RNA (transcriptome) to proteins (proteome) to metabolites (metabolome). It is important to note that proteins and metabolites can be modified, making them suitable targets for therapeutic interventions [7]. Proteomics and metabolomics were versatile analytical techniques that, in an ideal scenario, have the capability to comprehensively analyze the entire collection of proteins (proteome) or small molecules (metabolome) present in a biological sample [8]. Proteomics and metabolomics strategies have been widely employed to discover novel, sensitive, and specific biomarkers in transplantation [9]. Proteomics have been extensively employed to study the pathogenic mechanisms underlying PEDV infection [10–14], while the majority of these investigations have been conducted in vitro. Viral infections disrupt the metabolic processes of host cells, and the specific metabolic changes depend on the types of viruses and infected host cells. Viruses could trigger metabolic alterations to evade the immune response and enhance their replication within host cells, and glycolysis, lipid synthesis, tricarboxylic acid cycle, and nucleotide biosynthesis are the common metabolic processes hijacked by various viruses [15]. The hepatitis B virus utilizes glycolysis as a strategy to evade innate immune recognition, specifically by inhibiting the production of interferon induced by retinoic acid-inducible gene I [16]. Infection with the African swine fever virus led to an upregulation of the host tricarboxylic acid cycle and amino acid metabolism, which aided in promoting the replication of the virus within host cell [17]. PEDV infection leads to the impairment of intestinal cell function; however, the metabolic processes associated with PEDV infection and the specific key metabolites involved remain largely unexplored. At cell level, PEDV induced metabolic changes, and the serine, threonine, cysteine, and methionine metabolism were the most affected metabolic processes [18]. At the animal level, some mechanisms are still unknown.

In the present study, to identify proteins and metabolites related to the infective and resistance process of piglets infected with PEDV, we created the Yorkshire piglet model and performed jejunum proteomics and serum metabolomics. This study provides a novel perspective in advancing understanding of the pathogenesis of PEDV and new ideas for the prevention and control of coronavirus.

## 2. Materials and Methods

### 2.1. Animals and Samples

A total of 15 Yorkshire piglets involved in three litters were randomly selected from the Changping pig breeding farm of the Institute of Animal Science, Chinese Academy of Agricultural Sciences (Beijing, China). All 15 piglets were individually raised in a cardboard box of the animal rooms (Chinese Academy of Agricultural Sciences, Beijing, China) with strict control over pathogen intrusion and a constant temperature of 35 °C. These piglets did not take colostrum and were fed with the artificial milk (BaoBaoLe, Centree bio-tech (Wuhan) Co., Ltd., Wuhan, China) including $\geq$20% crude protein, $\leq$0.5% crude fiber, 0.5–1.1% calcium, $\geq$0.4% total phosphorus, $\leq$9% coarse ash, 0.4–1.5% sodium chloride and $\geq$1.6% lysine. Out of them, 10 individuals were infected with 1 mL PEDV at three days of age, and the remaining 5, as the control group, were administered 1 mL of 0.9% normal saline (Beijing Epsilon Biotechnology Co., Ltd., Beijing, China), in which the actual PEDV titer of challenge dose was 10 $TCID_{50}$ (50% tissue culture infective dose)/mL. $TCID_{50}$ was used for estimating the virus content. The PEDV (G2 mutant virus) was isolated from the jejunum of pigs in Beijing Liuma Pig Raising Technology Co., Ltd. in 2014 (Shunyi, Beijing, China), and has been used for infected animal experiments [19].

At 12 h after infection, we collected the feces of each pig and tested the PED positivity (viral load > $10^5$ $TCID_{50}$/mL) with the Anigen Rapid PED Ag Test Kit (Anigen, Republic of Korea). Within the next four days, five piglets died on Day 6 to Day 7 and were classified as the susceptible group. The remaining five piglets were distributed to the resistance group. For piglets of the control and resistance groups, we injected xylazine hydrochloride (0.2 mL/kg; Beijing Lab Anim Tech Develp Co., Ltd., Beijing, China) into each individual for anesthesia and dissection on Day 7. For all the piglets, we collected the serum and

tissues (duodenum, jejunum, ileum, colon, cecum and rectum) samples for follow-up research. The specific research design is shown in Figure 1.

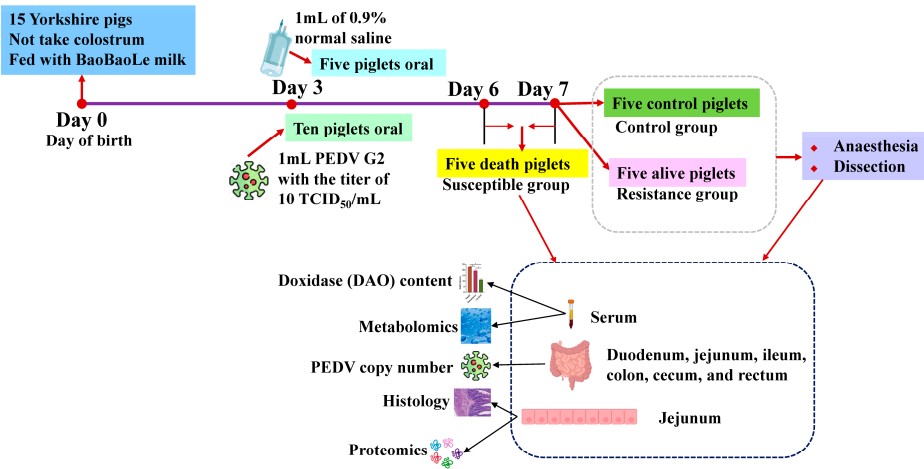

**Figure 1.** Research design of this study. The blood and tissue samples of susceptible group were collected when the piglets died on Day 6–7. The blood and tissue samples of control and resistance groups were collected after anesthesia and dissection on Day 7.

## 2.2. Measurements of PEDV and Degree of Jejunal Injury

A total of six bowel segments (duodenum, jejunum, ileum, colon, cecum and rectum) were collected to detect the PEDV copy number. We extracted the viral nucleic acid of each intestine using the MagaBio plus Viral DNA/RNA Purification Kit (Hangzhou Biotechnology Co., Ltd., Hangzhou, China), and reverse transcribed them to the cDNA using the TaKaRa Reverse Transcription Kit (TaKaRa Bio, Shiga, Japan). We designed the N gene primers (Supplementary Table S1) of the PEDV using Primer 3 [20] and synthesized them in Shenzhen Huada Gene Co., Ltd. (Shenzhen, China). The qRT-PCR amplifications were carried out on an Applied Biosystems PCR instrument (Thermo Fisher Scientific, Waltham, MA, USA) with the following procedure: 30 s at 95 °C, 40 cycles of 5 s at 95 °C and 34 s at 60 °C, 15 s at 95 °C, 1 min at 60 °C, 95 °C for 15 s. The PEDV copy number was obtained by substituting the Ct value into the standard curve constructed in our lab.

Enzyme diamine oxidase (DAO) was the gastrointestinal degradation of histamine [21], and its levels could be used to monitor the function of the small intestinal mucosal barrier [22]. We measured the DAO content in serum to assess for intestinal damage using the DAO ELISA kit (s51-96T, Boibio, Beijing, China). For the PEDV copy number, DAO content, length of villi, and weight change in piglets, we used the $t$-test to analyze the significance ($p < 0.05$).

## 2.3. Jejunum Tissue Histology

The 1 cm × 1 cm × 1 cm jejunum tissue of each piglet was fixed in 4% paraformaldehyde (Beijing Biotopped Technology Co., Ltd., Beijing, China), and was conducted in the paraffin sections as per the reported procedure [23]. In short, the fixed jejunum was cut, and the tissue slide was placed in a staining rack for conducting hematoxylin–eosin staining. Last, we used the Leica DM300 microscope (Leica Microsystems, Wetzlar, Germany) for imaging.

## 2.4. Protein Extraction of Jejunum and Four-Dimensional (4D) Data-Independent Acquisition (DIA)-Based Proteomics

About 40 μL uL of nanomagnetic beads was absorbed and washed once with a certain amount of wash buffer; then, equal volumes of wash buffer and jejunum were added and turned 360 degrees in a rotary mixer and incubated at 37 °C for 1 h. The supernatant was removed by magnetic separation and vibration, washing the beads for 5 min with

3 times the volume of wash buffer. The cleaning was repeated 3 times, and the enriched jejunum protein was obtained on the magnetic beads. In addition, the BSA standard protein solution was prepared according to the instructions of Bradford protein quantitative kit, in which gradient concentration ranged from 0 to 0.5 g/L. BSA standard protein and sample solutions with different dilution multiples were, respectively, added into a 96-well plate to fill the volume to 20 µL, and each gradient was repeated three times. The plate was quickly filled with 180 µL G250 dye solution and placed at room temperature for 5 min, and then the absorbance at 595 nm was detected. Finally, the standard curve was drawn with the absorbance of the standard protein solution and the protein concentration of the sample was calculated. Also, 20 µg of the protein sample was loaded to 12% sodium dodecyl sulfate–polyacrylamide gel electrophoresis (SDS-PAGE) to measure the protein quality at 80 V for 20 min. After the treatment of trypsin, each protein sample was performed using 4D-DIA liquid chromatography with tandem mass spectrometry (LC-MS/MS) analysis with the UltiMate 3000 (Thermo Fisher Scientific, Waltham, MA, USA) liquid chromatography system. The 4D-DIA-based proteomics is a developing technology that has high precursor identification specificity [24]. In the process, samples were reconstituted in 0.1% formic acid (FA), and 200 ng peptide was separated using an AUR3-15075C18 column (15 cm length, 75 µm i.d, 1.7 µm particle size, 120 A pore size, and IonOptics) with a 60 min gradient starting at 4% buffer B (80% acetonitrile with 0.1% FA) followed by a stepwise increase to 28% in 25 min, 44% in 10 min, 90% in 10 min, remaining there for 7 min, then equilibrated at 4% for 8 min. DIA data were acquired in DIA parallel accumulation serial fragmentation (PASEF) mode, and the precursor isolation window was set to 40 Da with a scanning range of 349 to 1229 $m/z$. During PASEF MS/MS scanning, the collision energy was ramped linearly as a function of the mobility from 59 eV (1/K0 = 1.6 Vs/cm$^2$) to 20 eV (1/K0 = 0.6 Vs/cm$^2$).

### 2.5. Metabolite Extraction of Serum and Ultrahigh-Performance Liquid Chromatography-MS/MS (UHPLC–MS/MS) Analysis

A total of 100 µL serum of each piglet was obtained from the blood by centrifuging at 2000 rpm for 10 min at 4 °C, and was then stored at −80 °C for metabolite extraction. All the serum samples were placed in Eppendorf (EP) tubes and resuspended with prechilled 80% methanol by well vortex. Then, the samples were incubated on ice for 5 min and centrifuged 20 min at 15,000× $g$ (4 °C). Some of supernatant was diluted to 53% methanol using LC-MS-grade water. The samples were subsequently transferred to a fresh EP tube and then were centrifuged for 20 min at 15,000× $g$ (4 °C). Finally, the supernatant was injected into the LC-MS/MS system for analysis.

UHPLC-MS/MS analyses were performed using a Thermo Syncronis C18 (2.1 mm × 100 mm, 1.7 µm) UHPLC system (ThermoFisher, Dreieich, Germany) coupled with an Orbitrap Q Exactive$^{TM}$ series mass spectrometer (ThermoFisher, Dreieich, Germany) at Scale Co., Ltd. (Beijing, China). Samples were injected onto a Hypesil Gold column (2.1 mm × 100 mm, 1.7 µm) with an 18 min linear gradient at a flow rate of 0.2 mL/min. The eluents A and B were 0.1% FA in water and acetonitrile, respectively. The solvent gradient was set as follows: 95% A for 0~1 min, 95~40% A for 1~5 min, 40~0% A for 5~8 min, 0% A for 8~11 min, 0~40% A for 11~14 min, 40~95% A for 11~15 min and 95% A for 15~18 min. Q Exactive$^{TM}$ series mass spectrometer was operated in positive/negative polarity mode with a spray voltage of 3.2 kV, capillary temperature of 320 °C, sheath gas flow rate of 40 arb and aux gas flow rate of 10 arb.

### 2.6. Identification of Proteins and Metabolites

Raw data of DIA were processed and analyzed using a Spectronaut 17 (Biognosys AG, Schlieren, Switzerland) with default settings to generate an initial target list. Data extraction was determined using Spectronaut according to the extensive mass calibration. Spectronaut was used to determine the ideal extraction window dynamically depending on item response theory calibration and gradient stability. Q-value (FDR) cutoff on precursor

level was 1% and protein level was 1%. Decoy generation was set to mutate, which was similar to scrambled but only applied a random number of assessment area position swamps (min = 2, max = length/2). Peptides that passed the 1% Q-value cutoff were used to calculate the major group quantities with MaxLF Q method.

The raw data files generated using UHPLC-MS/MS were processed using TraceFinder3.2.0 (ThermoFisher) to perform peak alignment, peak picking and quantitation for each metabolite. The main parameters were set to retention time tolerance of 0.2 min, actual mass tolerance of 5 ppm, signal/noise ratio of 3 and minimum intensity. After that, peak intensities were normalized to the total spectral intensity, and then peaks were matched with the mzCloud (https://www.mzcloud.org/, accessed on 21 September 2023) (URL) and self-built database to obtain accurate qualitative and relative quantitative results.

### 2.7. Data Analysis and Functional Annotation

For the identified proteins, we performed principal component analysis (PCA) for all identified proteins and metabolites using an R package to explore the repeatability and difference across the three groups. The differential analyses were conducted using a *t*-test across the three groups, namely, control vs susceptible, control vs resistance and resistance vs susceptible. When the *p* values of proteins were less than 0.05 and fold changes (FCs) of proteins were >2 or <0.5, these proteins were differentially expressed proteins (DEPs). The functional annotation of DEPs was conducted using KOBAS (http://bioinfo.org/kobas/, accessed on 21 September 2023) [25], and the enrichment with the corrected *p*-value < 0.05 was significant. In detail, we chose species Sus scrofa (pig) and entered the DEP lists to conduct annotation analysis. For metabolites, we performed partial least squares discriminant analysis (PLS-DA) to obtain the importance of projection (VIP) values and also performed the *t*-test across the three groups to identify the differentially accumulated metabolites (DAMs). The metabolites with *p* values less than 0.05, FC > 2 or <0.5, and VIP value > 1 were the DAMs. For the DEPs and DAMs, the volcano plots of proteins and metabolites were performed using ggplot2 in R according to log2 (fold change) and −log10 (*p*-value).

Weighted gene co-expression network analysis (WGCNA) is a tool used to analyze the association of genes with traits and find functional key modules [26]. We carried out WGCNA for the identified proteins and metabolites to discover the complex relationships between related modules and different piglet groups. The module with *p* < 0.01 and | correlation coefficient | > 0.6 was significant. Further, we used the DEPs and promising DAMs involved in significant modules of WGCNA to perform short timeseries expression miner (STEM) analysis [27] for identification of significant group features. For the promising DAMs involved in profiles with significant group features, we performed random forest analysis and used "Mean Decrease Accuracy" and "Mean Decrease Gini" to measure the importance of a metabolite in a random forest discriminative grouping. The metabolite with larger "Mean Decrease Accuracy" and "Mean Decrease Gini" values was more important in the random forest. For the key metabolites, we performed the functional annotation by MateboAnalyst 5.0 (https://www.metaboanalyst.ca/MetaboAnalyst/ModuleView.xhtml, accessed on 21 September 2023) [28].

In addition, we conducted an integrative analysis of DEPs and DAMs for each comparison group using the Spearman method [29], in which the correlations with correlation coefficient values > 0.8 or <−0.8, and *p* < 0.05 were selected as the significant integration.

### 3. Results

### 3.1. Determinations of PEDV Copy Numbers, Degree of Jejunum Damage and Weight Loss

We calculated the copy numbers of PEDV in duodenum, jejunum, ileum, colon, cecum and rectum of Yorkshire pigs, and found that the virus copy numbers in these six tissues of the susceptible group were higher than in those of the resistance group. The jejunum, ileum and colon of the susceptible group had significantly higher PEDV copy numbers than those of the resistance group (*p* < 0.05) (Figure 2A), and the virus copy numbers of

jejunum were the highest among these six bowel segments (Figure 2A). In addition, we performed qRT-PCR for the six bowel segments of the control piglets to measure their virus copy numbers, and no Ct value was found, which implied no PEDV. These results implied that the resistance group might have the ability to resist viral replication.

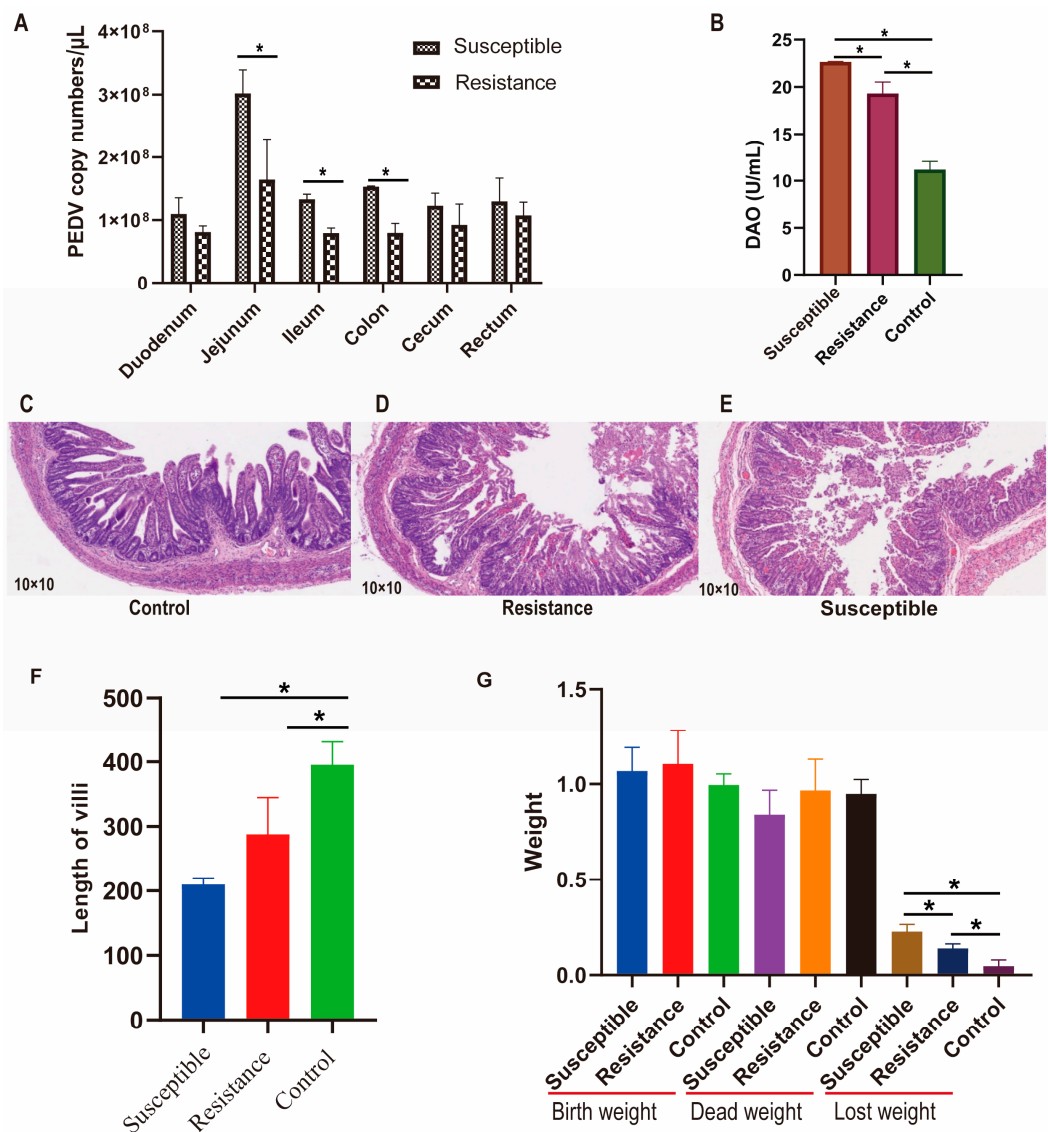

**Figure 2.** PEDV copy number, intestinal lesions and changes in weight. (**A**) PEDV copy numbers in six small intestine tissues. (**B**) Diamine oxidase (DAO) content in serum. (**C**–**E**) Paraffin sections (10 × 10) of jejunal tissues. (**F**) Length of villi. (**G**) Weight of piglets. Lost weight indicates a change between birth and death weight. * indicated $p < 0.05$.

The DAO content in serum was detected, and the result is displayed in Figure 2B. The DAO content in infection group was significantly higher than that in the control group, and the susceptible piglets had the highest content (Figure 2B). In addition, we performed paraffin sectioning of jejunal tissues (Figure 2C–F) and found that the susceptible and resistant piglets had an obvious loss of intestinal villi compared to the control piglets ($p < 0.05$). The susceptible group showed the worst intestinal injury with a high number of lost intestinal villi and congested interstitial vessels. In addition, we weighed the piglets at birth and death. There was no difference in their birth weight and dead weight, while their lost weight showed significant differences between any two groups ($p < 0.05$)

(Figure 2G). Hence, mortality in susceptible pigs might be due to severe intestinal damage and weight loss.

### *3.2. Protein Identification from the Jejunum of Piglets Involved in PEDV Model*

Based on the 4D-DIA quantification, a total of 7680 proteins were identified in all samples from susceptible, resistance and control groups (Supplementary Table S2). In addition, 7465 proteins were expressed in more than 50% of samples (≥7), and the following analyses were based on these 7465 proteins (Supplementary Table S3). We performed the PCA of these 7465 proteins across the three groups (Figure 3) and found that the proteins in the susceptible, resistance and control groups were clearly distinct from one another, indicating significant variations in proteomic profiles between any two groups.

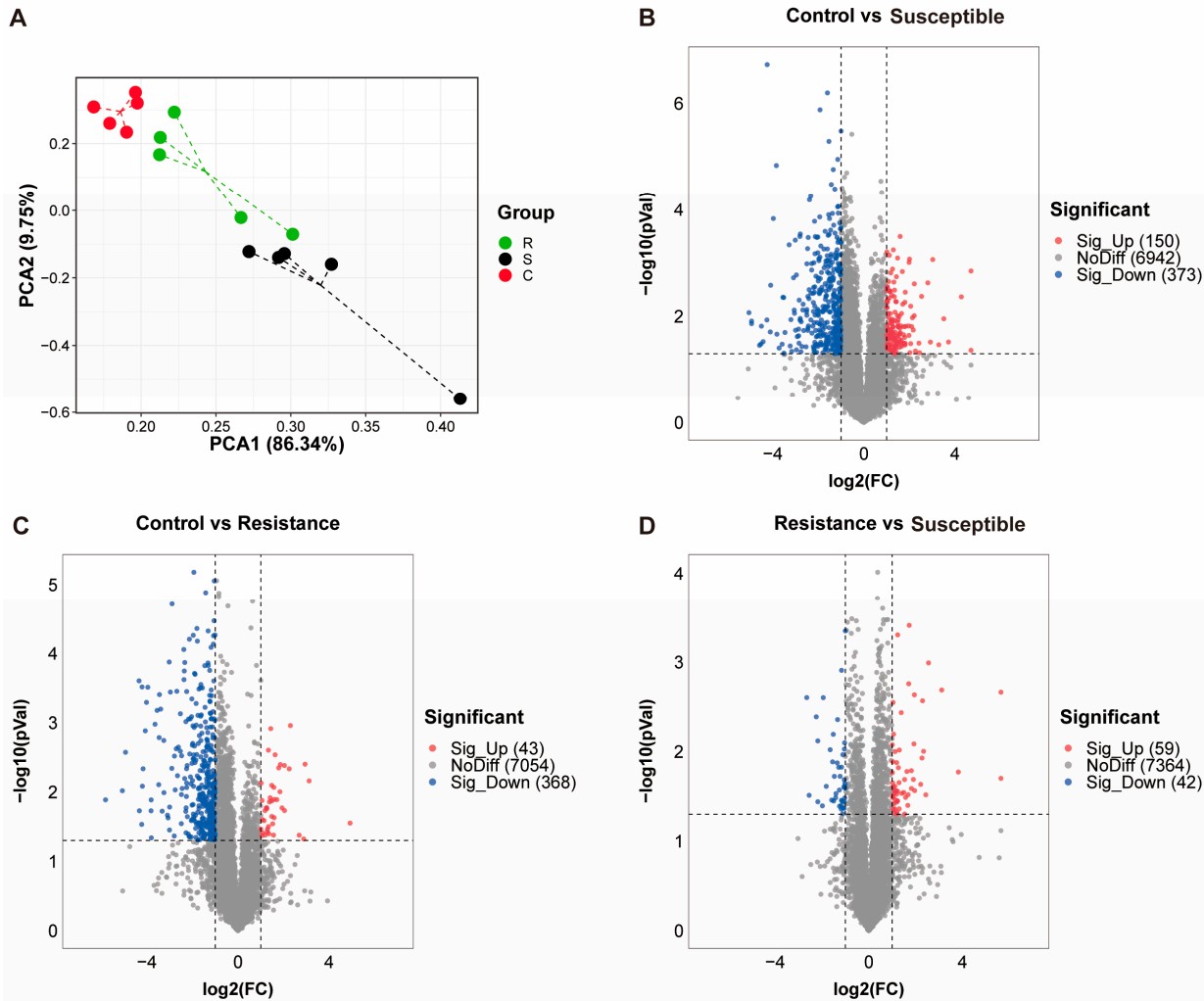

**Figure 3.** Principal component analysis (PCA) and difference analysis of proteins. (**A**–**D**) are the PCA and difference analysis results of the proteins, respectively. R: resistance. S: susceptible. C: control. Sig_up: significantly upregulated ($p < 0.05$ and fold change (FC) > 2). NoDiff: not different. Sig_Down: significantly downregulated ($p < 0.05$ and FC < 0.5). The numbers in brackets were the numbers of proteins.

### *3.3. Differential Expression Analysis and Functional Annotation of Protein*

After the differential analysis, the statistically significant difference ($p < 0.01$, and FC > 2 or <0.5) showed 734 DEPs, involving in control vs susceptible group (150 up-expressed and 373 down-expressed), control vs resistance group (43 up-expressed and 368 down-expressed) and resistance vs susceptible group (59 up-expressed and 42 down-

expressed) (Figure 3 and Supplementary Table S4). Most differences were measured between the control and infectious piglets, while the resistance vs susceptible had fewer DEPs. These showed that a large number of proteins were participated in the PEDV reaction. Figure 4A showed the overlap of DEPs from any comparison group, in which six proteins (A0A5G2QRW3, RHAG, ANK1, SRGN, CDV3 and SLC4A1) were the overlaps of three comparison groups.

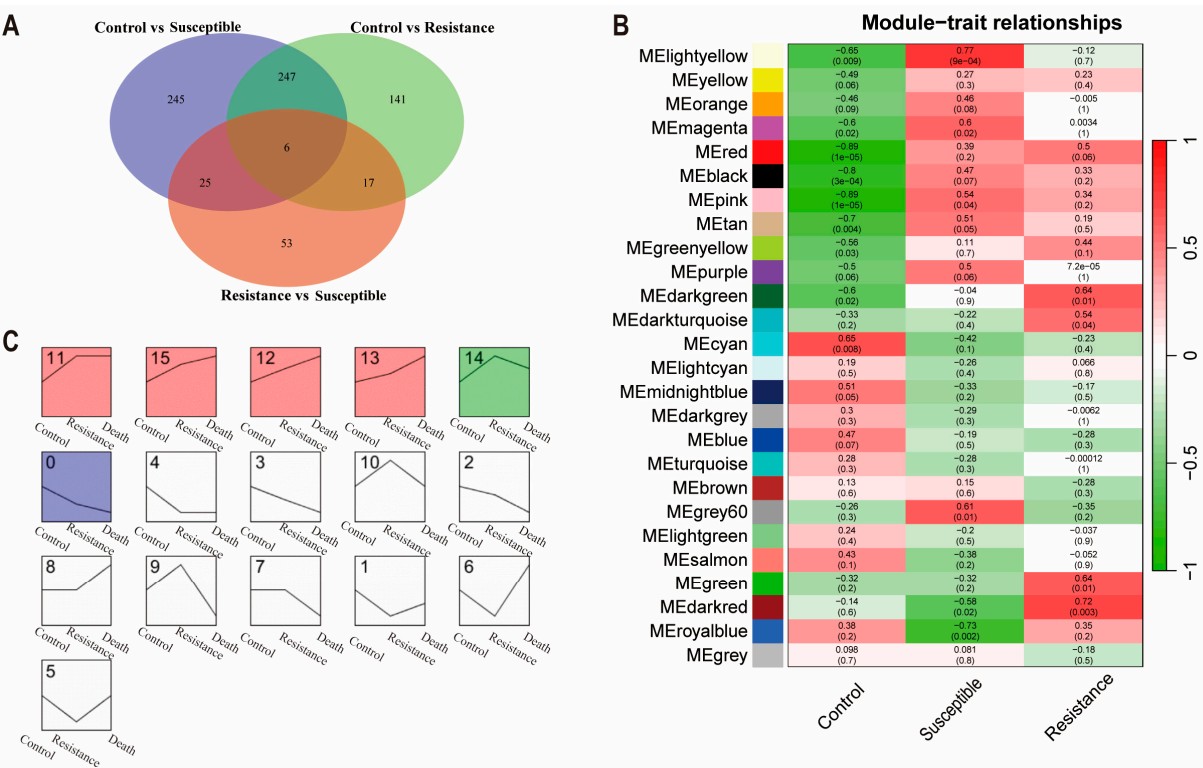

**Figure 4.** (**A**) Venn diagram of differentially expressed proteins. (**B**) Weighted gene co-expression network analysis (WGCNA) of proteins. The first row of numbers in the square denotes the relationships between proteins with each group, and the threshold is |correlation coefficient| > 0.6. The larger the number indicated, the higher the correlation. The second row of numbers in the square is the *p* value, and the threshold is *p* < 0.01. The smaller the value indicated, the greater significance. The modules with *p* < 0.01 and the |correlation coefficient| > 0.6 is significant. (**C**) Short timeseries expression miner (STEM) analysis results of proteins. The number in the module is the module name. The color map shows significant profiles, while the white does not. Death: susceptible group.

Then, the database of KOBAS was used for the functional annotation of DEPs, of which the enrichment with the corrected *p*-value < 0.05 was significant. The up- and downregulated DEPs in the control vs susceptible group were, respectively, involved in 75 (20 KEGG pathways and 55 GO terms) and 55 (5 KEGG pathways and 50 GO terms) significant enrichments (Supplementary Table S5). The up- and downregulated DEPs in the control vs resistance group were separately involved in 68 (10 KEGG pathways and 58 GO terms) and 21 (GO terms) significant enrichments (Supplementary Table S5). For the resistance vs susceptible group, the upregulated DEPs were enriched in 4 GO terms (Supplementary Table S5), and the downregulated DEPs participated in three KEGG pathways and 103 GO terms (Supplementary Table S5). According to the functional annotation, we found that the upregulated proteins in the control group were mainly associated with some metabolic pathways, such as glycosaminoglycan degradation, amino sugar and nucleotide sugar metabolism, and fatty acid degradation. Also, these upregulated proteins in the control group were strongly involved in the PPAR signaling pathway, tight junction, apoptosis, renin–angiotensin system and platinum drug resistance, which impact

the normal immunity of the individual. The upregulated proteins in the resistance group were mainly involved in the GO terms related to immunity, such as T cell co-stimulation, activation of NF-κB-inducing kinase activity, apoptotic signaling pathway and defense response to Gram-positive bacterium. The upregulated proteins in susceptible group largely had relations with the body damage, such as platelet aggregation, blood coagulation and negative regulation of growth (Supplementary Table S5). Also, some disease resistance and signal transduction were identified in the upregulated proteins of the susceptible group, such as the p53 signaling pathway, complement and coagulation cascades, RNA transport, and endocytosis (Supplementary Table S5). However, in these disease resistance and signal transduction pathways, the number of proteins was small, which meant little contribution and possibly the main reason for the susceptibility of piglets due to their weak disease resistance.

### 3.4. Metabolomic Profiling of Jejunum from Piglets Involved in PEDV Model

Metabolite levels were examined in the susceptible, resistance and control piglets with five biological replicates within each group. With the UHPLC-MS/MS, we identified 719 annotated metabolites for each individual (Supplementary Table S3). We conducted the PCA of the identified metabolites across the three groups (Figure 5). Noteworthily, metabolites in susceptible, resistance and control groups were clearly separated from each other, demonstrating the existence of major metabolomic differences between any two groups.

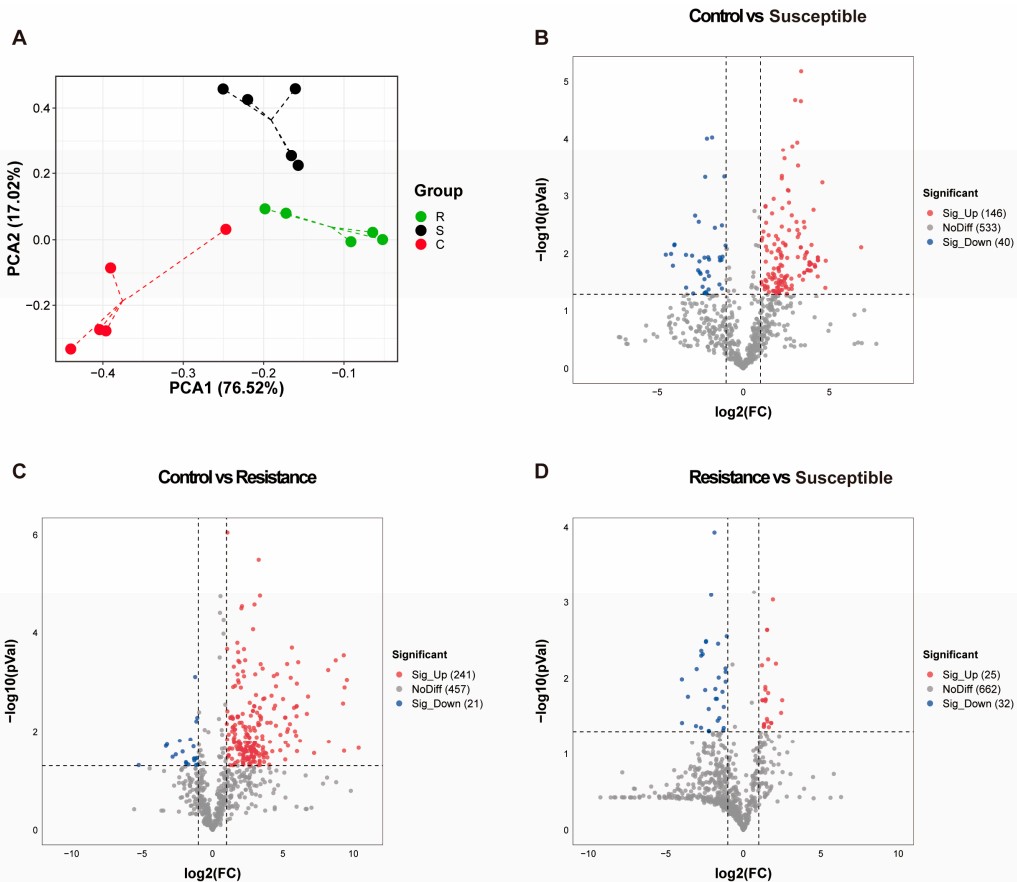

**Figure 5.** Principal component analysis (PCA) and difference analysis of metabolites. (**A**–**D**) are the PCA and difference analysis results of the proteins, respectively. R: resistance. S: susceptible. C: control. Sig_Up: significant up-accumulated ($p < 0.05$ and fold change (FC) > 2). NoDiff: not different. Sig_Down: significant down-accumulated ($p < 0.05$ and fold change (FC) < 0.5). The numbers in brackets are the numbers of metabolites.

Differential accumulation analysis was performed, and the metabolite with the *p*-value of < 0.05 and FC > 2 or <0.5 was the differentially accumulated metabolite (DAM). In total, 330 DAMs were identified (Figure 5 and Supplementary Table S4), including 186 in the control vs susceptible group (146 up-accumulated and 40 down-accumulated), 262 in the control vs resistance group (241 up-accumulated and 21 down-accumulated) and 57 in the resistance vs susceptible group (25 up-accumulated and 32 down-accumulated).

### 3.5. Biomarker Metabolites Identification and Functional Annotation

PLS-DA (partial least squares discriminant analysis) was used to detect the metabolites that contributed the most to group segregation, known as variable importance in projection (VIP) scores. A total of 376 metabolites with a VIP score of > 1 were considered highly influenced (Supplementary Table S6). To explore the candidate biomarker metabolites that could distinguish the piglets involved in susceptible, resistance and control, 208 metabolites as significant DAMs (FC > 2 or FC < 0.5, *p* < 0.05 and VIP score > 1) were identified (Supplementary Table S6), containing 112 in the control vs susceptible group (11 up-accumulated and 2 down-accumulated), 145 in the control vs resistance group (42 up-accumulated and 23 down-accumulated) and 47 in the resistance vs susceptible group (84 up-accumulated and 27 down-accumulated).

For the significant DAMs, we determined the functional annotation and a total of 46 metabolites were discovered with corresponding functional annotation (Supplementary Table S5). The results displayed that the significantly high-accumulated metabolites in control group had associations with metabolic regulation, such as glycerophospholipid metabolism, biosynthesis of secondary metabolites, biosynthesis of amino acids, protein digestion and absorption, and primary bile acid biosynthesis. The significantly high-accumulated metabolites in the resistance group were also involved in metabolic pathways, such as phenylalanine metabolism, tryptophan metabolism, biosynthesis of secondary metabolites, microbial metabolism in diverse environments, degradation of aromatic compounds, and nicotinate and nicotinamide metabolism. The significantly high-accumulated metabolites in the susceptible group were related to metabolism and some signal transduction pathways, for example, the cAMP signaling pathway and HIF-1 signaling pathway.

### 3.6. Weighted Gene Co-Expression Network Analysis (WGCNA) and Short Timeseries Expression Miner (STEM) Analysis

A protein co-expression network was constructed using WGCNA, and 26 modules were obtained Figure 4B. Six modules (light yellow, red, black, pink, tan and cyan) were significantly associated with the control group (*p* < 0.01 and the |correlation coefficient| > 0.6) (Figure 4B). The modules, light yellow, grey60, and royal blue, were significantly associated with susceptible piglets (*p* < 0.01 and the |correlation coefficient| > 0.6) (Figure 4B). Additionally, modules, dark green, green, and dark red, were found to be strongly relevant to resistant piglets (*p* < 0.05 and the |correlation coefficient| > 0.6) (Figure 4B). In the significant modules, 552 proteins were DEPs, and we used them to perform STEM analysis. In this study, six profiles (11, 15, 12, 13, 14 and 0) including 408 proteins had the significant group feature (Figure 4C). The proteins in profiles 11, 15, 12 and 13 had the highest expressions in the susceptible group, and the proteins in profiles 14 and 0, respectively, had the highest expressions in the resistance and susceptible groups (Figure 4C). The proteins involved in significant modules and profiles are shown in Supplementary Table S7.

For the metabolites, we also conducted WGCNA and found six significant modules (*p* < 0.01 and the |correlation coefficient| > 0.6) (Figure 6A), namely, blue, magenta, purple and red modules associated with the control group; black module associated with the resistance group; and tan modules related to the susceptible group. These significant modules included 339 metabolites, of which 157 metabolites were the candidate biomarker metabolites (Supplementary Table S7). Further, we performed STEM analysis for the 157 candidate biomarker metabolites and discovered two profiles (4 and 1) with the significant group

feature (Figure 6B), which contained 122 metabolites (Supplementary Table S7). These metabolites were mainly PC, LysoPC and SM classes.

**Figure 6.** (**A**) Weighted gene co-expression network analysis (WGCNA) of metabolites. The first row of numbers in the square denotes the relationships between metabolites and group. Second row of numbers in the square denotes the *p* value. The modules with *p* < 0.01 and the |correlation coefficient| > 0.6 are significant. (**B**) Short timeseries expression miner (STEM) analysis results of metabolites. The color map shows significant profiles, while the white does not. Death: susceptible group. (**C**) Thirty metabolites with significant features using random forest analysis. C: control group. S: susceptible group. R: resistance group.

## 3.7. Detection of Characteristic Biomarkers for Susceptible and Resistance

Some metabolites and proteins had significant differences in resistance vs susceptible and control vs susceptible comparisons, while having no significance in control vs resistance comparison. Also, some metabolites and proteins were strongly changed in resistance vs susceptible and control vs resistance comparisons but showed no significance in control vs susceptible comparisons. Such metabolites and proteins included in the significant modules of WGCNA were considered as characteristic biomarkers for susceptible and resistance groups. In total, we found 12 metabolites (succinic acid, D-(+)-Malic acid, malic acid, isocitric acid, 2,6-dihydroxypurine, D-Arabitol, adonitol, propionic acid, fumaric acid, 9(10)-DiHOME, (9Z)-(7S,8S)-dihydroxyoctadecenoic acid and (9R,10R)-dihydroxyoctadecanoic acid) and eight proteins (LOC100523310, FOXM1, FHL2, MTRR, SPTLC3, ENPEP, HACD1 and FMO1) as the characteristic biomarkers for susceptibility (Table 1). For resistance, we proposed 12 proteins (TP53 regulated inhibitor of apoptosis 1, ABHD3, IRS2, IFITM1, RYR2, C2H19orf25, LOC100153925, FLVCR1, GPX2, GPSM3, TSPO and CD180) as the characteristic biomarkers (Table 1).

**Table 1.** The characteristic biomarkers for susceptibility and resistance.

| Characteristic Biomarkers for Susceptibility | | | Characteristic Biomarkers for Resistance | |
| --- | --- | --- | --- | --- |
| **Metabolites** | **Protein Descriptions** | **Proteins (Gene Name)** | **Protein Descriptions** | **Proteins (Gene Name)** |
| Succinic acid | GB1/RHD3-type G domain-containing protein | LOC100523310 | TP53 regulated inhibitor of apoptosis 1 | |
| D-(+)-malic acid | Forkhead box M1 | FOXM1 | Abhydrolase domain containing 3, phospholipase | ABHD3 |
| Malic acid | Four and a half LIM domains 2 | FHL2 | Insulin receptor substrate 2 | IRS2 |
| Isocitric acid | 5-methyltetrahydrofolate-homocysteine methyltransferase reductase | MTRR | Interferon-induced transmembrane protein 1 | IFITM1 |
| 2,6-dihydroxypurine | Serine palmitoyltransferase 3 | SPTLC3 | Ryanodine receptor 2 | RYR2 |
| D-arabitol | Glutamyl aminopeptidase | ENPEP | Chromosome 2 C19orf25 homolog | C2H19orf25 |
| Adonitol | Very-long-chain (3R)-3-hydroxyacyl-CoA dehydratase | HACD1 | Mitotic spindle-organizing protein 2A isoform X4 | LOC100153925 |
| Propionic acid | Flavin-containing monooxygenase 1 | FMO1 | FLVCR heme transporter 1 | FLVCR1 |
| Fumaric acid | | | Glutathione peroxidase | GPX2 |
| 9(10)-DiHOME | | | G protein signaling modulator 3 | GPSM3 |
| (9Z)-(7S,8S)-Dihydroxyoctadecenoic acid | | | Translocator protein | TSPO |
| (9R,10R)-Dihydroxyoctadecanoic acid | | | CD180 antigen | CD180 |

### 3.8. Random Forest Analysis of the Key Metabolites

Using random forest analysis, the top 30 metabolites with MeanDecreaseAccuracy > 0.003361 were detected (Figure 6C), of which, 10 (histamine, PC(18:3e/2:0), LysoPC(20:3(5Z,8Z,11Z)), PC(14:1e/3:0), LysoPC(20:4(8Z,11Z,14Z,17Z)), PC(18:4e/2:0), Linolelaidic Acid (C18:2N6T), LysoPC(20:2(11Z,14Z)), LysoPC(22:5(7Z,10Z,13Z,16Z,19Z)), and PC(o-16:1(9Z)/20:4(8Z,11Z, 14Z,17Z))) and 12 metabolites (PC(18:3e/22:3), PC(14:0/22:1(13Z)), D-(+)-Proline, D-Phenyl-alanine, SM(d18:0/16:1(9Z)), 47-diphenyl_110_phenanthroline, L-Phenylalanine, N-_3-Oxohexanoyl_homoserine lactone, SM(d18:1/16:0), penicillamine, PC(18:2(9Z,12Z)/P-18:1(9Z)), and palmitoyl sphingomyelin), respectively, had the highest accumulations in resistant and susceptible piglets, and SM(d18:1/24:0), 4-Methoxycinnamaldehyde, SM(d18:0/24:1(15Z)), SM(d18:0/22:0), PC(16:0/17:2), PC(18:1e/18:3), PC(14:1(9Z)/P-18:1(11Z)) and PC(18:2/19:2) had the highest accumulations in control group (Figure 6C). These 30 metabolites might be the key promising biomarker metabolites for the corresponding group.

### 3.9. Features of the Protein–Metabolite Interactome

To construct a comprehensive profile of piglets infected with PEDV and identify the relationships between proteins and metabolites, multi-omics analysis integrating proteomic and untargeted metabolic data based on the same biological samples was conducted. Significantly positive and negative correlations between proteins and metabolites were identified (Supplementary Table S8). For the DEPs and promising DAMs in comparison of control vs susceptible, a total of 3430 significant correlations were detected, in which 337 proteins and 102 metabolites were included. For comparisons of control vs resistance and resistance vs susceptible, 3073 (296 proteins and 142 metabo-

lites) and 102 (23 and 24 metabolites) significant correlations were, respectively, identified. In the resistance vs susceptible comparison, six PC ((PC(16:2e/2:0), PC(18:4e/2:0), PC(14:1e/2:0), PC(18:3e/2:0), PC(14:0e/3:0), and PC(18:5e/4:0))) and LysoPC metabolites (LysoPC(18:2(9Z,12Z), LysoPC(20:4(8Z,11Z,14Z,17Z)), LysoPC(16:1(9Z)), LysoPC(20:3(5Z,8Z, 11Z)), LysoPC(17:0), and LysoPC(22:5(7Z,10Z,13Z,16Z,19Z))) were negatively correlated to SPTB, A0A5G2QRW3, FLACC1, ATP5MJ, LOC110259958, RHAG, ANK1, SRGN, CDV3, SLC4A1 and HBB proteins, in which these proteins had the functions of cell development and oxygen metabolism, such as erythrocyte development, granzyme-mediated apoptotic signaling pathway, oxygen carrier activity and oxygen transport. D-Sorbitol, isocitric acid, citraconic acid, nicotinuric acid, glycodeoxycholic acid, D-(+)-Malic acid, malic acid and fumaric acid were positively correlated to SPTB, TARBP2, A0A5G2QRW3, FGG, FLACC1, SLC25A25, LOC110259958 and FGB proteins, which were involved in some metabolic and immune pathways, such as metabolic pathways and complement and coagulation cascades (Figure 7A–C).

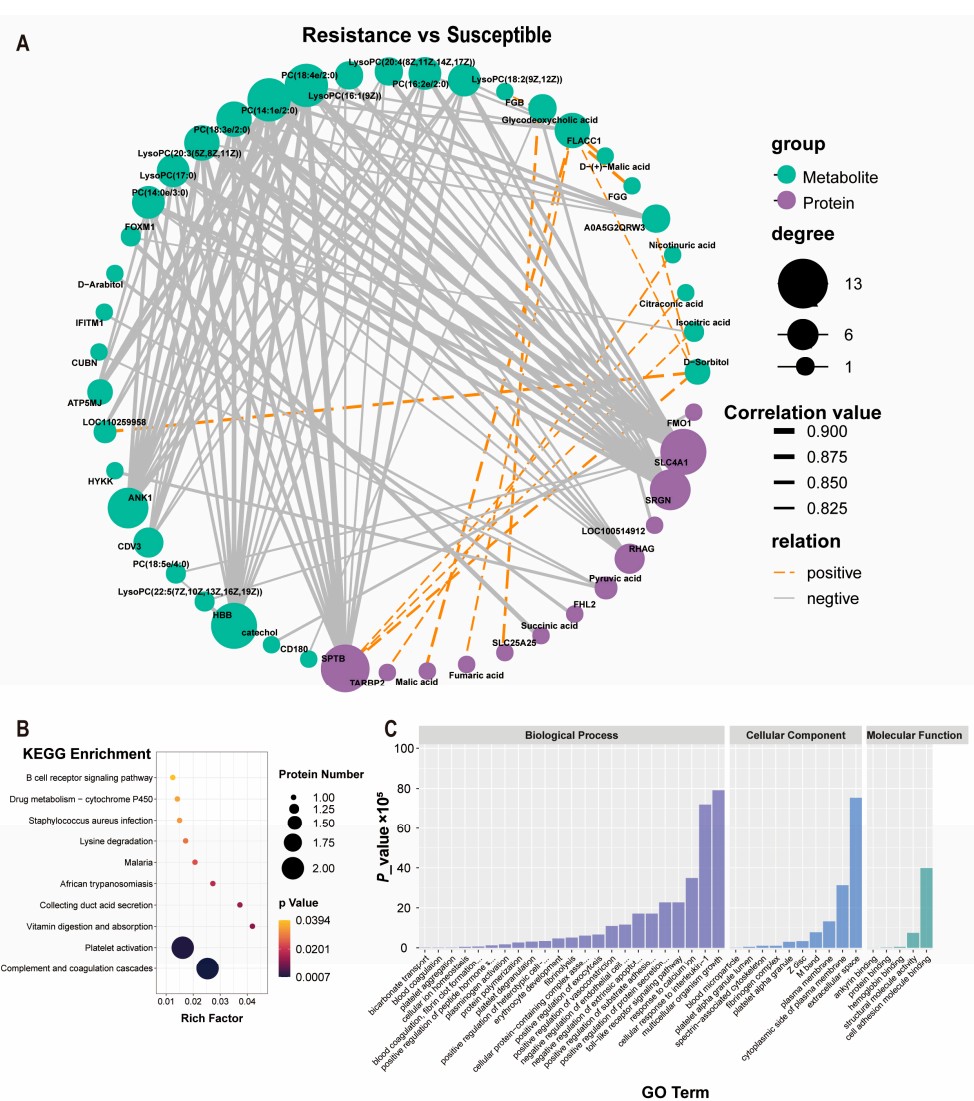

**Figure 7.** (**A**) Significant interactions between key proteins and metabolites involved in resistance vs susceptible. (**B**) Main KEGG pathways of the proteins in the significant interactions. (**C**) Main GO terms of the proteins in the significant interactions.

## 4. Discussion

In this study, we constructed the Yorkshire piglet model infected with PEDV, and investigated the jejunal proteomics and serum metabolomics. The piglets used in this study

did not take colostrum, and no virus was in the memory of the immune system. Therefore, phenotypic and mechanistic changes could reflect the process of PEDV infection. The phenotype and PCA results revealed clear differences across the susceptible, resistance, and control groups. We provided key proteins and metabolites that might be involved in PEDV response, which might provide new insights and markers for preventing the spread of PEDV.

Here, the phenotypic data demonstrated that the dead piglets had the most severe intestinal damage and PEDV copy numbers, which might be the cause of susceptibility. Here, we measured the copy numbers of PEDV in duodenum, jejunum, ileum, colon, cecum and rectum, and found that the virus copy numbers of jejunum were highest among these six bowel segments (Figure 2A). Between susceptible and resistance groups, the copy number of PEDV in jejunum was most significant among these six bowel segments (Figure 2A). This indicates that the PEDV mainly colonized the jejunum, which is consistent with previous reports [2,30]. Therefore, we selected the jejunum for further proteomics. In addition, it is not clear why the jejunum is the main pathogenic site of PEDV; hence, it is necessary to study the relevant PEDV mechanisms in jejunum. Metabolite profiles of serum are important indicators of physiological and pathological states and have been reported to help understand the mechanism of disease occurrence and progression on the metabolic level [31]. In this study, we also conducted the serum metabolomics for three groups.

Our proteomics data revealed that large amounts of proteins were DEPs between the control and PEDV infection groups, which implies that they were mobilized when the piglets were infected with PEDV. The upregulated expressed proteins in the control and PEDV-infected piglets were mainly involved in metabolic and signal transduction pathways, respectively. The results indicate that individual metabolic regulation was very important in resistance to PEDV. These DEPs might promote virus survival and reproduction or help the body destroy viruses, and their specific function needs to be further researched for this to be determined. The infection of PEDV has important associations with apoptosis [32], and NF-κB has been reported to be suppressed by PEDV [33]. Dendritic cells play a key role in activating the intestinal immune response, and could promote the proliferation and differentiation of T cells, which are antigen-presenting cells of the mammalian immune system, which could fight PEDV by secreting proinflammatory cytokines [34]. Hence, for the comparison group of resistance vs susceptible, the significant terms involved in DEPs with high expressions in resistance piglets were immunity, which might be associated with resistance to PEDV. The upregulated expressed DEPs in dead piglets were mainly involved in body damage, which implies that PEDV caused severe injury and susceptibility for these individuals. For example, RNA transport controls over-localized protein synthesis and cell polarity [35], and endocytosis impacts autophagy, metabolism, cell division, apoptosis, cellular defense and intestinal permeabilization [36].

The metabolomics data provided 157 promising metabolites using difference analysis and WGCNA, and the further STEM and random forest analysis provided 30 key promising biomarker metabolites for the corresponding group. Histamine plays a crucial role in the development of various allergic diseases [37] and impacts mucosal immune homeostasis within the gastrointestinal tract [38]. Histamine also has associations with DAO [39,40]. In this study, the DAO had significant changes in different groups, and greater histamine content in the control group might maintain mucosal metabolism. A total of 11 PC metabolites were detected as the key promising biomarker metabolites, which were the most abundant phospholipids in all mammalian cell membranes and showed importance of phospholipid metabolism in regulating lipid, lipoprotein and whole-body energy metabolism [41]. SMs are the main sources of drugs, cosmetics and health products [42], and serve the crucial purpose of protecting against several pathogens [43]. SM(d18:0/16:1(9Z)) and SM(d18:1/16:0), with lower accumulations in resistance and susceptible groups, might play an important role in resistance to PEDV. LysoPC has been proposed to activate cells from a number of lineages [44], and has interactions with the metastatic potential of tumor cells [45]. The present study also revealed

that four LysoPC metabolites (LysoPC(20:3(5Z,8Z,11Z)), LysoPC(20:4(8Z,11Z,14Z,17Z)), LysoPC(20:2(11Z,14Z)) and LysoPC(22:5(7Z,10Z,13Z,16Z,19Z))), respectively, had the highest and lowest accumulations in resistance and susceptible groups, indicating their functions in resistance to PEDV. In addition, when the piglets were infected with PEDV, the contents of 4-Methoxycinnamaldehyde changed little, and the accumulations of D-(+)-Proline, D-Phenylalanine, 47-diphenyl_110_phenanthroline, L-Phenylalanine, N-_3-Oxohexanoyl_homoserine lacto-ne, linolelaidic acid (C18:2N6T), penicillamine and palmitoyl sphingomyelin increased, which implies that these metabolites might be markers of pathogenesis. Among these metabolites, some of them have been reported to impact immunity, for example, LysoPC induced humoral and cellular immune responses [46], and penicillamine D-Phenylalanine was related with mortality [47].

According to the difference analysis and WGCNA, we detected 12 and 8 proteins as characteristic biomarkers for resistance and susceptibility, respectively. The characteristic biomarkers for resistance were mainly involved in fighting PEDV and regulating body homeostasis. ABDH3 is the site for phospholipids, which are crucial cellular and intracellular compounds and are required for active transport and cell signaling [48]. IRS2 is a critical node of insulin signaling [49], and acts as a protective factor [50]. IFITM1 is associated with antiviral activity of PEDV [51], and FLVCR1 is the receptor of pathogenic subgroup C feline leukemia virus [52]. RYR2 is responsible for releasing $Ca^{2+}$ from the sarcoplasmic reticulum of cardiomyocytes, whose aberrant activity is life-threatening [53]. GPX2 is associated with TGEV-induced apoptosis in IPEC-J2 cells [54], GPSM3 has the immune features of regulatory T cells, neutrophils and a lower proportion of monocytes [55]. The high expression of TSPO is related with low injury [56], and CD180 is involved in innate immune defense [57]. TP53 regulates inhibition of apoptosis 1-modulated apoptotic pathways, preventing the formation of the apoptosome complex [58], which might fight PEDV by slowing down apoptosis. For C2H19orf25 and LOC100153925, there are no related reports, which might also be characteristic biomarkers for resistance based on our results. The characteristic biomarkers for susceptibility identified in this study are mainly involved in immune regulation. FOXM1 plays a critical role in inflammation and disease, and its deregulation likely leads to an imbalance in immune responses [59]. FHL2 is the NF-κB inhibitor, suggesting an ineffective "cytokine storm" [60]. MTRR is one of the key regulatory enzymes in the folate metabolic pathway, while intestinal folate metabolism disorder might result in systemic deficiency [61]. SPTLC3 is functionally involved in ceramide biosynthesis and is related to metabolic disorders [62]. HACD1 in mammalian has been linked to certain muscle diseases [63], while HACD1 and LOC100523310 have not been reported as being related with virus immunity. ENPEP is a blood pressure regulator [64], and FMO1 is a biomarker of some diseases, such as nonalcoholic fatty liver disease [65] and peritoneal metastasis in gastric cancer [66]. In addition, we propose phenotypic indicators of susceptibility after PEDV infection, which provides a resource for further understanding of the PEDV infection mechanism.

Until now, studies on PEDV have mainly concerned cellular transcriptomics, proteomics and metabolomics [18,67], which are limited in representing the reality of individual responses. In addition, little information has been made available to characterize the jejunum proteome and metabolome of piglets infected with PEDV, especially for resistant piglets. Our study conducted a multi-omics analysis of metabolomics and proteomics datasets based on the same biological samples and conducted the interactome of protein–metabolite to present the significant relationship pairs associated with PEDV responses for piglets within seven days of age. These networks elucidated the biological pathway from gene encoding to the production of final metabolic products and uncovered significant drivers linked to PEDV pathogenesis. Our research results indicate that the resistant pigs were able to show resistance to PEDV because their intestinal damage had not yet reached extreme levels, so they might have the corresponding regulatory mechanisms in their bodies. In three comparison groups, a large number of significant correlations were detected, and these characteristics could potentially be utilized as candidate biomarkers for diagnosis

and targets for therapeutic intervention. In the present study, a series of relationships between proteins and metabolites were also discovered, and these are believed to play an important role in maintaining host health or as the marks for indicating infection, resistance or susceptibility.

## 5. Conclusions

In summary, this study showed an integrative analysis of proteomic and metabolomic data based on the same biological samples from piglets infected with PEDV and control individuals. A total of 20 and 12 characteristic biomarkers for susceptible and resistant piglets infected with PEDV were detected, and their functions were analyzed. The complex network of characteristic proteins and metabolites depict the crucial immune metabolism linked to PDEV and indicate the PEDV response mechanism of key molecules. These findings provide molecular resources for piglet resistance to PEDV.

**Supplementary Materials:** The following supporting information can be downloaded at: https://www.mdpi.com/article/10.3390/agriculture14010043/s1, Table S1: Primer information of N gene of PEDV; Table S2: Proteins identified in this study; Table S3: Proteins and metabolites used for the analysis in this study; Table S4: Differentially expressed proteins (DEPs) and differentially accumulated metabolites (DAMs) between any two groups; Table S5: Functional annotation of differentially expressed proteins (DEPs) and key differentially accumulated metabolites (DAMs); Table S6: Metabolites with VIP > 1; Table S7 Important proteins and metabolites identified using WGCNA, STEM and forest analyses; Table S8: Features of the protein–metabolite interactome.

**Author Contributions:** L.W. and L.S. designed the experiments. L.S., H.L. and C.Z. collected the samples. L.S. analyzed the data and wrote the manuscript. All authors have read and agreed to the published version of the manuscript.

**Funding:** This work was supported by the National Key R&D Program of China (2021YFD1301101), Chinese Academy of Agricultural Sciences Foundation (2023-YWF-ZYSQ-03), Agricultural Science and Technology Innovation Program (ASTIP-IAS02), and National Swine Industry Technology System (CARS-35).

**Institutional Review Board Statement:** The animals used in this study were approved by the Animal Ethics Committee of the Institute of Animal Science, Chinese Academy of Agricultural Sciences (Beijing, China).

**Data Availability Statement:** Data are available in the Supplementary Material.

**Acknowledgments:** We are especially grateful to Meimei Zhang and Na Yuan (Beijing Vica Biotechnology Co., Ltd.) for providing the PEDV.

**Conflicts of Interest:** The authors declare no conflict of interest.

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
