# Peer review of "Proteomic and Metabolomic Profiling Elucidate the Impact of PEDV on Yorkshire Piglets and Reveal the Underlying Molecular Mechanism of PEDV Response"

_agriculture, doi:10.3390/agriculture14010043_

Round 1
Reviewer 1 Report
Comments and Suggestions for Authors The manuscript is generally well-written, but attention should be paid to improving the clarity of certain sections, especially in the introduction and discussion.Clarify the methods employed for proteomic and metabolomic profiling. Provide a more comprehensive explanation of data preprocessing and quality control steps. Additionally, elaborate on statistical methods used for data analysis, including multiple testing corrections.
The presentation of proteomic and metabolomic data is comprehensive. However, figures and tables could benefit from clearer labeling and more detailed legends. Consider providing additional context for key findings to aid in interpretation.
Statistical Significance:
Ensure that statistical significance is appropriately indicated in the results section. Consider presenting p-values and effect sizes for key comparisons.
The authors do well in connecting the results to the study objectives. However, a more in-depth discussion of the biological relevance of identified proteins and metabolites in the context of PEDV response is warranted.
Comparison to Previous Studies:
Expand on the comparison of your findings with existing literature. Highlight similarities and differences and discuss potential reasons for any discrepancies.
The conclusion should succinctly summarize the main findings and their implications. Clearly state how your study contributes to our understanding of the molecular mechanisms underlying PEDV response in Yorkshire piglets.
A final proofread for grammar and style is recommended. Additionally, ensure consistency in terminology throughout the manuscript. Comments on the Quality of English Language
Minor editing of English language required
Author Response
Re: agriculture-2701930
Dear Reviewer 1,
Thank you very much for providing an opportunity for us to revise our paper for your journal. We also wish to thank you for the positive comments and constructive suggestions on the manuscript. We have made the essential revisions suggested as follows.
Comments and Suggestions for Authors
The manuscript is generally well-written, but attention should be paid to improving the clarity of certain sections, especially in the introduction and discussion.
Q1: Clarify the methods employed for proteomic and metabolomic profiling. Provide a more comprehensive explanation of data preprocessing and quality control steps. Additionally, elaborate on statistical methods used for data analysis, including multiple testing corrections.
Response 1: Thanks so much for your kind suggestions. As suggested, we have provided the comprehensive explanation of data preprocessing and quality control in the Materials and methods as follows:
“For the PEDV copy number, DAO content, length of villi, and wight change of piglets, we used T-test to analyze the significance (P < 0.05).” Please see the lines 120-122, page 3.
“The differential analyses were conducted by T-test across the three groups, namely, control vs susceptible, control vs resistance, and resistance vs susceptible. When the P values of proteins were less than 0.05 and fold changes (FCs) of proteins were > 2 or < 0.5, these proteins were differentially expressed proteins (DEPs).”. Please see the lines 201-205, page 5.
“For metabolites, we did the partial least squares discriminant analysis (PLS-DA) to obtain the importance of projection (VIP) values and also did the T-test across the three groups to identify the differentially accumulated metabolites (DAMs). The metabolites with P values less than 0.05, FC > 2 or < 0.5, and VIP value > 1 were the DAMs.”. Please see the lines 208-210, page 5.
“Raw Data of DIA was processed and analyzed by Spectronaut 17 (Biognosys AG, Switzerland) with default settings to generate an initial target list. Data extraction was determined by Spectronaut according to the extensive mass calibration. Spectronaut determined the ideal extraction window dynamically depending on item response theory calibration and gradient stability. Q-value (FDR) cutoff on precursor level was 1% and protein level was 1%. Decoy generation was set to mutate, which was similar to scrambled but only applied a random number of assessment area position swamps (min = 2, max = length/2). Peptides which passed the 1% Q-value cutoff were used to calculate the major group quantities with MaxLF Q method.”. Please see the lines 182-190, page 5.
“The raw data files generated by UHPLC-MS/MS were processed using the Trace-Finder3.2.0 (ThermoFisher) to perform peak alignment, peak picking, and quantitation for each metabolite. The main parameters were set to retention time tolerance of 0.2 minutes, actual mass tolerance of 5ppm, signal/noise ratio of 3, and minimum intensity. After that, peak intensities were normalized to the total spectral intensity, and then peaks were matched with the mzCloud (https://www.mzcloud.org/) and self-built da-tabase to obtain accurate qualitative and relative quantitative results.”. Please see the lines 191-197, page 5.
Q2: The presentation of proteomic and metabolomic data is comprehensive. However, figures and tables could benefit from clearer labeling and more detailed legends. Consider providing additional context for key findings to aid in interpretation.
Statistical Significance: Ensure that statistical significance is appropriately indicated in the results section. Consider presenting p-values and effect sizes for key comparisons.
Response 2: Thanks so much for your kind suggestions. As suggested, we have added the clearer labeling and more detailed legends as follows:
“Figure 2. PEDV copy number, intestinal lesions, and changes of weight. A The PEDV copy numbers in six small intestine tissues. B The diamine oxidase (DAO) content in serum. C-E The paraffin sections (10×10) of jejunal tissues. F The length of villi. G The weight of piglets. Lose weight indicated the change between birth and death weight. * indicated P < 0.05.”. Please see the lines 243-246, page 6.
“Figure 3. Principal component analysis (PCA) and difference analysis of proteins. A and B-D were the PCA and difference analysis results of the proteins, respectively. R: Resistance. S: Suscep-tible. C: Control. Sig_up: significant up-regulated (P < 0.05 and fold change (FC) > 2). NoDiff: not different. Sig_Down: significant down-regulated (P < 0.05 and FC < 0.5). The numbers in brackets were the numbers of proteins.”. Please see the lines 267-271, page 7.
“Figure 4. A Venn of differentially expressed proteins. B Weighted gene co-expression network analysis (WGCNA) of proteins. The first row of numbers in a square was the relationships between proteins with each group, and the threshold was |correlation coefficient| > 0.6. The larger the number indicated the higher the correlation. The second row of numbers in a square was the P value, and the threshold was P < 0.01. The smaller the value indicated the greater significance. The modules with P < 0.01 and the |correlation coefficient| > 0.6 were significant. C Short time-series expression miner (STEM) analysis results of proteins. The number on the module was the module name. The color map is significant profiles, while the white was not.”. Please see the lines 283-290, page 8.
“Figure 5. Principal component analysis (PCA) and difference analysis of metabolites. A and B-D were the PCA and difference analysis results of the proteins, respectively. R: Resistance. S: Suscep-tible. C: Control. Sig_up: significant up-accumulated (P < 0.05 and fold change (FC) > 2). NoDiff: not different. Sig_Down: significant down-accumulated (P < 0.05 and fold change (FC) < 0.5). The numbers in brackets were the numbers of metabolites.”. Please see the lines xx, page xx.
“Figure 6. A Weighted gene co-expression network analysis (WGCNA) of metabolites. First row of numbers in a square was the relationships between metabolites with group. Second row of numbers in a square was the P value. The modules with P < 0.01 and the |correlation coefficient| > 0.6 were significant. B Short time-series expression miner (STEM) analysis results of metabolites. The The color map is significant profiles, while the white was not. C 30 metabolites with significant features by random forest analysis. C: Control group. S: Susceptible group. R: Resistance group.”. Please see the lines 325-329, page 9.
Q3: The authors do well in connecting the results to the study objectives. However, a more in-depth discussion of the biological relevance of identified proteins and metabolites in the context of PEDV response is warranted.
Response 3: Thanks so much for your kind suggestions. As suggested, we have added the discussion of the key proteins and metabolites as follows:
“IRS2 was a critical node of insulin signaling [49], and could act as a protective factor [50]. IFITM1 was associated with antiviral activity of PEDV [51], and FLVCR1 was the receptor of pathogenic subgroup C feline leukemia virus [52]. RYR2 was responsible for releasing Ca2+ from the sarcoplasmic reticulum of cardiomyocytes, whose aberrant ac-tivity engendered life-threatening [53]. GPX2 was associated with the TGEV-induced apoptosis in IPEC-J2 cells [54], GPSM3 had the immune features regulatory T cells, neutrophils, and a lower proportion of monocytes [55]. The high expression of TSPO had a relation with low injury [56], and CD180 was involved in innate immune defence [57]. TP53 regulated inhibitor of apoptosis 1 modulated apoptotic pathways preventing the formation of the apoptosome complex [58], which might fight PEDV by slowing down apoptosis. For C2H19orf25 and LOC100153925, there were no related reports, which might be also the characteristic biomarkers for resistance based on our results. The characteristic biomarkers for susceptibility identified in this study were mainly involved in immune regulation. FOXM1 played a critical role in inflammation and disease, and its deregulation likely led to the imbalance of immune responses [59]. FHL2 was the NF-κB inhibitor, suggesting the ineffective "cytokine storm" [60]. MTRR was one of the key regulatory enzymes in the folate metabolic pathway, while the in-testinal folate metabolism disorder might result in systemic deficiency [61]. SPTLC3 was functionally involved in ceramide biosynthesis and had relations with metabolic disorders [62]. HACD1 in mammalian had been linked to certain muscle diseases [63]. while HACD1 and LOC100523310 had not been reported to have relations with virus immunity. ENPEP was the blood pressure regulator [64], and FMO1 had been the bi-omarker of some diseases, such as nonalcoholic fatty liver disease [65] and peritoneal metastasis in gastric cancer [66]. In addition, we proposed phenotypic indicators of susceptibility after PEDV infection, which provided a resource for further under-standing of the PEDV infection mechanism.”. Please see the lines 524-548, page 16.
“Among these metabolites, some of them had been reported to impact the immunity, for example, LysoPC induced the humoral and cellular immune responses [46], and peni-cillamine D-Phenylalanine had relation with the mortality [47].”. Please see the lines 516-518, page 15.
Q4: Comparison to Previous Studies: Expand on the comparison of your findings with existing literature. Highlight similarities and differences and discuss potential reasons for any discrepancies.
Response 4: Thanks so much for your kind comments and suggestions. As suggested, we have added the discussion in the revised manuscript, namely, “Until now, studies on PEDV were mainly about cellular transcriptomics, proteomics, and metabolomics [18, 60], which were limited in representing the reality of individual responses. In addition, little information had been available to characterize the jejunum proteome and metabolome of piglets infected with PEDV, especially for the resistance piglets. Our study conducted the multi-omics analysis of metabolomics and proteomics datasets based on the same biological samples and conducted the interactome of protein-metabolite to present the significant relationship pairs associated with PEDV responses for the piglets within seven days of age.”. Please see the lines 549-555, page 16.
Q5: The conclusion should succinctly summarize the main findings and their implications. Clearly state how your study contributes to our understanding of the molecular mechanisms underlying PEDV response in Yorkshire piglets.
Response 5: Thanks so much for your kind suggestions. As suggested, we have revised the conclusion to “In summary, this study showed an integrative analysis of proteomic and metabolomic data based on the same biological samples from piglets infected with PEDV and control individuals. A total of 20 and 12 characteristic biomarkers for susceptible and resistance piglets infected with PEDV were detected, and their functions were analyzed. The complex network of characteristic proteins and metabolites depicted the crucial immune metabolism linked to PDEV, and indicated the PEDV response mechanism of key molecules. These findings provided molecular resources for piglet resistance to PEDV.”. Please see the lines 568-574, page 16.
Q6: A final proofread for grammar and style is recommended. Additionally, ensure consistency in terminology throughout the manuscript.
Response 6: Thanks so much for your kind suggestions. As suggested, we have carefully improved the English grammar, and those marked in yellow are the places modified. Please see the marked in yellow of revised manuscript.
In addition, we have ensured the consistency in terminology throughout the manuscript, such as “4DIA” to “4D-DIA”, and “NF-kappaB” to “NF-κB”. Please see the lines 149 and 480, pages 4 and 15.
Q7: Comments on the Quality of English Language: Minor editing of English language required
Response 7: Thanks so much for your kind suggestion. As suggested, we have carefully improved the English grammar, and those marked in yellow are the places modified. Please see the marked in yellow of revised manuscript.

Reviewer 2 Report
Comments and Suggestions for Authors
"Proteomic, and metabolomic profiling elucidate the impact of PEDV on Yorkshire piglets and reveal the underlying molecular mechanism of PEDV response" by She et al experimentally demonstrates using several assays to find differential protein and metabolite expression during infection of piglets resistant to PEDV or dying from the infection compared to the control without infection. Several observations were interestingly made. Here are certain points that must be addressed before publication in Agriculture.
Major points:
- Has the PEDV G2 mutant virus been described in any publications? If so, please cite. Provide more information about it? Why was the strain selected for these experiments?
- Principle and explanation of DIA readout must be described. At this stage, general readers are not able to read and understand what to expect from the analysis.
- The plausible reasons why jejunum was the more severely infected should be discussed. Especially, this data was not novel as it was published already in previous report [2].
- More explanation and discussion must be made on why the resistance piglets did not die even though they were severely and obviously showed the loss of intestinal villi.
- Regarding the PCA analysis, it would be more beneficial for readers if the all 'control', 'resistance', and 'death' groups were presented in the same PCA plots, both proteomics and metabolomics. The three-group analysis will let the readers judge how close and distinct between each group more efficiently.
- Many of the abbreviations in the paper were never mentioned or explained in full, especially, technical assays. It disservice the readers since it would not help explanation and make readers understand the intention, conclusion, and the points of using them.
- What are the Figure 4B and 4C telling the readers. Significance of those numbers has not explained.
- Results from GO term and KEGG analyses must be more discussed. Relate those results to the conclusion and explain why 1) the infected groups (resistance and death) share in common, which suggests the infection/response process and 2) resistance and death groups were biologically distinct/what were the determinant of the resistance that made the piglet survived.
- "The up-regulated proteins in death group largely had relations with some disease resistance and signal transduction" >> Should the "disease resistance" proteins down-regulated instead? That would easily explain how they died. Is there any other hypothesis could be made? Please discuss.
- The rational of studying jejunal proteomics is easily understandable. But the rationale for performing serum metabolomics that is relevant for viral infection needs to be further explained.
- Describe briefly how the 'functional annotation' was performed.
- Discuss what up-/down-regulated metabolites mean in terms of biology/viral infection and how piglets showed different phenotypes (resistance and death).
- During the Result part, it was difficult to read and follow when the authors continously showed a lot of numbers without explaining their significance. Though some of them were mentioned and discussed in the Discussion part, this should be to certain extent explained in the Result part, too, in order to let the readers understand what they intended and concluded.
- Lines 365, describe what 'Some phenotypes" mean in this context.
- Abbreviation and brief explanation of each blood routine indicators must be stated and explained.
- In order to understand the biology of this viral infection, it might be great if the degree of changes in proteins and metabolites stated, too, besides stating only that certain proteins/metabolites were up-/down-regulated.
- Take care of chemical names, such as D-_+_-Proline. There are many that should be taken care of.
- Raw data of proteomics and metabolomics should be deposited in a public server, too.
- Certain experimental validation of proteomics and metabolomics data should be done, too.
Minor points:
- Carefully check affiliations.
- What is BaoBaoLe milk? Describe the ingredients and/or content of it.
- What is TCID50? Explanation is required.
- Mention the dissection day (7) of the control group in the Materials and Methods, too.
- Citation for Primer 3 is needed.
- Check if the authors want to use 4D-DIA or 4DIA. NF-kappaB or NF-κB and so on. There were many. Please make it consistent.
- Please rewrite the first sentence(?) in Lines 365-366.
Comments on the Quality of English Language
N/A
Author Response
Re: agriculture-2701930
Dear Reviewer 2,
Thank you very much for providing an opportunity for us to revise our paper for your journal. We also wish to thank you for the positive comments and constructive suggestions on the manuscript. We have made the essential revisions suggested as follows.
Comments and Suggestions for Authors
"Proteomic, and metabolomic profiling elucidate the impact of PEDV on Yorkshire piglets and reveal the underlying molecular mechanism of PEDV response" by She et al experimentally demonstrates using several assays to find differential protein and metabolite expression during infection of piglets resistant to PEDV or dying from the infection compared to the control without infection. Several observations were interestingly made. Here are certain points that must be addressed before publication in Agriculture.
Major points:
Q1: - Has the PEDV G2 mutant virus been described in any publications? If so, please cite. Provide more information about it? Why was the strain selected for these experiments?
Response 1: Thanks so much for your kind comments and suggestions. The PEDV G2 mutant virus has been used in our previous study. As suggested, we have cited it and provided more information about it in the revised manuscript, namely, “The PEDV (G2 mutant virus) was isolated from the jejunum of pigs in Beijing Liuma Pig Raising Technology Co., LTD in 2014 (Shunyi, Beijing, China), and has been used for the infected animal experiments [19]”. Please see the lines 89-91, page 2.
Also, this virus was the popular strain, and it was isolated from the jejunum of diseased pigs, hence, we selected this virus for these experiments for more in line with the actual production, which had more practical value.
Q2: - Principle and explanation of DIA readout must be described. At this stage, general readers are not able to read and understand what to expect from the analysis.
Response 2: Thanks so much for your kind suggestion. As suggested, we have added the full name of 4D-DIA and its meaning, and we have added the relevant reference as follows:
“Protein extraction of jejunum and four-dimensional (4D)-data independent acquisition (DIA)-based proteomics”. Please see the lines 129-130, page 4.
“The 4D-DIA-based proteomics is a developing technology that has high precursor identification specificity [24]”. Please see the lines 149-150, page 4.
Q3: - The plausible reasons why jejunum was the more severely infected should be discussed. Especially, this data was not novel as it was published already in previous report [2].
Response 3: Thanks so much for your kind comment and suggestion. A large number of research reports have found that the jejunum is the main pathogenic site of PEDV, while the mechanism is unclear. As suggested, we have added the corresponding discussion in the revised manuscript, namely, “These indicated that the PEDV mainly colonized the jejunum, which was consistent with the previous reports [2, 30]. Therefore, we selected the jejunum for further proteomics and metabolomics. In addition, it is not clear why the jejunum is the main pathogenic site of PEDV, hence, it is necessary to study the relevant PEDV mechanisms in jejunum.”. Please see the lines 464-467, page 14.
Q4: - More explanation and discussion must be made on why the resistance piglets did not die even though they were severely and obviously showed the loss of intestinal villi.
Response 4: Thanks so much for your kind suggestion. As suggested, we have added the corresponding discussion in the revised manuscript as follows:
“Our proteomics data revealed that large amounts of proteins were DEPs between the control and PEDV infection groups, which implied that they were mobilized when the piglets were infected with PEDV. The up-regulated expressed proteins in control and PEDV-infected piglets were mainly involved in metabolic and signal transduction pathways, respectively. The results indicated that individual metabolic regulation was very important in resistance to PEDV. These DEPs might be promoting virus survival and reproduction or helping the body destroy viruses, and their specific function needed to be further researched to determine. The infection of PEDV had important associations with the apoptosis [32], and NF-κB had been reported to be suppressed by the PEDV [33]. Dendritic cells had the key role in activating the intestinal immune re-sponse, and could promote the proliferation and differentiation of T cells, which are an antigen-presenting cells of the mammalian immune system, which could fight the PEDV by secreting pro-inflammatory cytokines [34]. Hence, for the comparison group of resistance vs susceptible, the significant terms involved in DEPs with high expres-sions in resistance piglets were immunity, which might be associated with resistance to PEDV. The upregulated expressed DEPs in dead piglets were mainly involved in the body damages, which implied that the PEDV caused severe injury and susceptible for these individuals. For example, RNA transport controlled the over localized protein synthesis and cell polarity [35], and endocytosis impacted the autophagy, metabolism, cell division, apoptosis, cellular defense, and intestinal permeabilization [36].” Please see the lines 472-491, page 15.
Q5: - Regarding the PCA analysis, it would be more beneficial for readers if the all 'control', 'resistance', and 'death' groups were presented in the same PCA plots, both proteomics and metabolomics. The three-group analysis will let the readers judge how close and distinct between each group more efficiently.
Response 5: Thanks so much for your kind comment and suggestion. As suggested, we have added the PCA involved three groups in the revised manuscript as follows:
Figure 3. Principal component analysis (PCA) and difference analysis of proteins. A and B-D were the PCA and difference analysis results of the proteins, respectively. R: Resistance. S: Susceptible. C: Control. Sig_up: significant up-regulated. NoDiff: not different. Sig_Down: significant down-regulated. The numbers in brackets were the numbers of proteins.
Figure 5. Principal component analysis (PCA) and difference analysis of metabolites. A and B-D were the PCA and difference analysis results of the proteins, respectively. R: Resistance. S: Susceptible. C: Control. Sig_up: significant up-accumulated. NoDiff: not different. Sig_Down: significant down-accumulated. The numbers in brackets were the numbers of metabolites.
Q6: - Many of the abbreviations in the paper were never mentioned or explained in full, especially, technical assays. It disservice the readers since it would not help explanation and make readers understand the intention, conclusion, and the points of using them.
Response 6: Thanks so much for your kind comment and suggestion. As suggested, we have added the full names for all the abbreviations in the revised manuscript as follows:
- “weighted gene co-expression network analysis (WGCNA)”. Please see the lines 18-19, page 1.
- “short time-series expression miner (STEM)”. Please see the line 19, page 1.
- “hematoxylin-eosin”. Please see the line 127, page 3.
- “50% tissue culture infective dose (TCID50)”. Please see the line 88, page 2.
- “four-dimensional (4D)-data independent acquisition (DIA)”. Please see the lines 129-130, page 4.
- “sodium dodecyl sulfate–polyacrylamide gel electrophoresis (SDS-PAGE)”. Please see the line 145, page 4.
- “liquid Chromatography with tandem mass spectrometry (LC-MS/MS)”. Please see the lines 147-148, page 4.
- “formic acid (FA)”. Please see the line 151, page 4.
- “acetonitrile”. Please see the line 153, page 4.
- “DIA-parallel accumulation serial fragmentation (PASEF)”. Please see the lines 155-156, page 4.
- “ultra-high performance liquid chromatography-MS/MS (UHPLC–MS/MS)”. Please see the lines 160-161, page 4.
- “eppendorf (EP)”. Please see the line 164, page 4.
- “item response theory”. Please see the line 185, page 5.
- “assessment area”. Please see the lines xx, page xx.
- “importance in projection (VIP)”. Please see the line 188, page 5.
Q7: - What are the Figure 4B and 4C telling the readers. Significance of those numbers has not explained.
Response 7: Thanks so much for your kind comment. In Figure 4B, the first row of numbers in a square was the relationships between proteins with each group, and the threshold was |correlation coefficient| > 0.6. The larger the number indicated the higher the correlation. The second row of numbers in a square was the P value, and the threshold was P < 0.01. The smaller the value indicated the greater significance.
In Figure 4C, the number on the module was the module name.
As suggested, we have added the detailed description, namely, “Figure 4. A Venn of differentially expressed proteins. B Weighted gene co-expression network analysis (WGCNA) of proteins. The first row of numbers in a square was the relationships between proteins with each group, and the threshold was |correlation coefficient| > 0.6. The larger the number indicated the higher the correlation. The second row of numbers in a square was the P value, and the threshold was P < 0.01. The smaller the value indicated the greater significance. C Short time-series expression miner (STEM) analysis results of proteins. The number on the module was the module name. The color map is significant profiles, while the white was not.” Please see the lines 283-290, page 8.
Q8: - Results from GO term and KEGG analyses must be more discussed. Relate those results to the conclusion and explain why 1) the infected groups (resistance and death) share in common, which suggests the infection/response process and 2) resistance and death groups were biologically distinct/what were the determinant of the resistance that made the piglet survived.
Response 8: Thanks so much for your kind suggestions. Pigs infected with PEDV will mobilize their immune systems, even if they eventually die. Hence, it is reasonable that there were common enrichments both resistance and susceptible groups. Actually, the up-regulated proteins in susceptible group largely had relations with the body damage, such as platelet aggregation, blood coagulation, and negative regulation of growth, which was the leading cause of death.
As suggested, we have added these in the revised manuscript, as follows:
“The up-regulated proteins in susceptible group largely had relations with the body damage, such as platelet aggregation, blood coagulation, and negative regulation of growth (Table S5). Also, some disease resistance and signal transduction were identified in the up-regulated proteins of susceptible group, such as p53 signaling pathway, complement and coagulation cascades, RNA transport, and endocytosis (Table S5). While, in these disease resistance and signal transduction pathways, the number of proteins was few, which meant little contribution and might be the main reason for the susceptibility of piglets due to their weak disease resistance.”. Please see the lines 308-316, pages 8-9.
“The infection of PEDV had important associations with the apoptosis [32], and NF-κB had been reported to be suppressed by the PEDV [33]. Dendritic cells had the key role in activating the intestinal immune response, and could promote the proliferation and differentiation of T cells, which are an antigen-presenting cells of the mammalian immune system, which could fight the PEDV by secreting pro-inflammatory cytokines [34]. Hence, for the comparison group of resistance vs susceptible, the significant terms involved in DEPs with high expressions in resistance piglets were immunity, which might be associated with resistance to PEDV. The upregulated expressed DEPs in dead piglets were mainly involved in the body damages, which implied that the PEDV caused severe injury and susceptible for these individuals. For example, RNA transport controlled the over localized protein synthesis and cell polarity [35], and endocytosis impacted the autophagy, metabolism, cell division, apoptosis, cellular defense, and intestinal permeabilization [36].”. Please see the lines 479-491, page 15.
Q9: - "The up-regulated proteins in death group largely had relations with some disease resistance and signal transduction" >> Should the "disease resistance" proteins down-regulated instead? That would easily explain how they died. Is there any other hypothesis could be made? Please discuss.
Response 9: Thanks so much for your kind comments and suggestions. Pigs infected with PEDV will mobilize their immune systems, even if they eventually die. Hence, it is reasonable that these up-regulated proteins in death group are involved in disease resistance. Also, the results of this study illustrated it. While the up-regulated proteins in susceptible group largely had relations with the body damage, such as platelet aggregation, blood coagulation, and negative regulation of growth, which was the leading cause of death.
As suggested, we have more clearly stated the Results and Discussion sections in the revised manuscript as follows:
“The up-regulated proteins in susceptible group largely had relations with the body damage, such as platelet aggregation, blood coagulation, and negative regulation of growth (Table S5). Also, some disease resistance and signal transduction were identified in the up-regulated proteins of susceptible group, such as p53 signaling pathway, complement and coagulation cascades, RNA transport, and endocytosis (Table S5). While, in these disease resistance and signal transduction pathways, the number of proteins was few, which meant little contribution and might be the main reason for the susceptibility of piglets due to their weak disease resistance.”. Please see the lines 308-316, pages 8-9.
“The infection of PEDV had important associations with the apoptosis [32], and NF-κB had been reported to be suppressed by the PEDV [33]. Dendritic cells had the key role in activating the intestinal immune response, and could promote the proliferation and differentiation of T cells, which are an antigen-presenting cells of the mammalian immune system, which could fight the PEDV by secreting pro-inflammatory cytokines [34]. Hence, for the comparison group of resistance vs susceptible, the significant terms involved in DEPs with high expressions in resistance piglets were immunity, which might be associated with resistance to PEDV. The upregulated expressed DEPs in dead piglets were mainly involved in the body damages, which implied that the PEDV caused severe injury and susceptible for these individuals. For example, RNA transport controlled the over localized protein synthesis and cell polarity [35], and endocytosis impacted the autophagy, metabolism, cell division, apoptosis, cellular defense, and intestinal permeabilization [36].”. Please see the lines 479-491, page 15.
Q10: - The rational of studying jejunal proteomics is easily understandable. But the rationale for performing serum metabolomics that is relevant for viral infection needs to be further explained.
Response 10: Thanks so much for your kind comment. Metabolite profiles of serum can be regarded as important indicators of physiological and pathological states and may aid understanding of the mechanism of disease occurrence and progression on the metabolic level, and provide information enabling identification of early and differential metabolic markers of disease.
As suggested, we have added the corresponding discussion in the revised manuscript, namely, “Metabolite profiles of serum are important indicators of physiological and patholog-ical states and have been reported to help understand the mechanism of disease oc-currence and progression on the metabolic level [31]. In this study, we also conducted the serum metabolomics for three groups.” Please see the lines 467-471, pages 14-15.
Q11: - Describe briefly how the 'functional annotation' was performed.
Response 11: Thanks so much for your kind suggestion. We used the KOBAS to annotate the function of the DEPs, and the website provided the steps. As suggested, we have added the corresponding methods in the revised manuscript, namely, “The functional annotation of DEPs was conducted by the KOBAS (http://bioinfo.org/kobas/) [25], and the enrichment with the corrected P-value < 0.05 was significant. In detail, we chose species of Sus scrofa (pig) and put the DEP lists to conduct the annotation analysis.” Please see the lines 205-208, page 5.
Q12: - Discuss what up-/down-regulated metabolites mean in terms of biology/viral infection and how piglets showed different phenotypes (resistance and death).
Response 12: Thanks so much for your kind suggestions. As suggested, we have added the discussion in the revised manuscript as follows:
“Among these metabolites, some of them had been reported to impact the immunity, for example, LysoPC induced the humoral and cellular immune responses [46], and penicillamine D-Phenylalanine had relation with the mortality [47].”. Please see the lines 516-518, page 15.
Q13: - During the Result part, it was difficult to read and follow when the authors continously showed a lot of numbers without explaining their significance. Though some of them were mentioned and discussed in the Discussion part, this should be to certain extent explained in the Result part, too, in order to let the readers understand what they intended and concluded.
Response 13: Thanks so much for your kind suggestions. As suggested, we have added these contents in the revised manuscript as follows:
“These results implied that the resistance group might have the ability to resist viral replication.”. Please see the lines 240-241, page 6.
“Hence, mortality in susceptible pigs might be due to severe intestinal damage and weight loss.”. Please see the lines 256-257, page 7.
“These showed that a large number of proteins were participated in the PEDV reaction.”. Please see the lines 278-279, pages 7-8.
Q14: - Lines 365, describe what 'Some phenotypes" mean in this context.
Response 14: Thanks so much for your kind comment and suggestion. The phenotypes were indicated metabolites. As suggested, we have revised the “phenotypes” to “metabolites”. Please see the lines 392 and 394, page 11.
Q15: - Abbreviation and brief explanation of each blood routine indicators must be stated and explained.
Response 15: Thanks so much for your kind suggestion. We measured some routine blood indicators, while the results were not particularly good. Hence, we deleted this section content.
Q16: - In order to understand the biology of this viral infection, it might be great if the degree of changes in proteins and metabolites stated, too, besides stating only that certain proteins/metabolites were up-/down-regulated.
Response 16: Thanks so much for your kind comments and suggestions. In this study, we used P < 0.05 and fold change (FC) > 2 or < 0.5 to identify the differentially expressed proteins (DEPs), and used P < 0.05, fold change (FC) > 2 or < 0.5, and importance of projection (VIP) > 1 to detect the differentially accumulated metabolites (DAMs). Hence, when we mentioned the up-/down-regulated proteins/metabolites, their changes were more than two folds. Since there were many DEPs and DAMs, the respective change folds were not added one by one.
Q17: - Take care of chemical names, such as D-_+_-Proline. There are many that should be taken care of.
Response 17: Thanks so much for your kind suggestions. As suggested, we have carefully checked these, and revised the “D-_+_-Proline” to “D-(+)-Proline”, and “Ca2+” to “Ca2+”. Please see the lines 512-513 and 526, pages 15 and 16.
Q18: - Raw data of proteomics and metabolomics should be deposited in a public server, too.
Response 18: Thanks so much for your kind suggestion. The raw data including protein expression and metabolite content were shown in the Table S3.
Q19: - Certain experimental validation of proteomics and metabolomics data should be done, too.
Response 19: Thanks so much for your kind suggestions. We are very agreed with the comment and suggestion of the review, namely, certain experimental validation was needed. While, this article was mainly aimed at full-spectrum research on the proteome and metabolome for the piglets infected PEDV, and provided data resources to interested researchers. The verification of key proteins and metabolites would be carried out as a separate project in the future.
Minor points:
Q1: - Carefully check affiliations.
Response 1: Thanks so much for your kind suggestion. As suggested, we have confirmed the affiliations, namely, “Lijun Shi 1, Huihui Li 1, Chunxiang Zhou 2 and Lixian Wang 1,*
1 Institute of Animal Science, Chinese Academy of Agricultural Sciences, Beijing 100193, China
2 Huanghe Science and Technology College”. Please see the line 7, page 1.
Q2: - What is BaoBaoLe milk? Describe the ingredients and/or content of it.
Response 2: Thanks so much for your kind comment and suggestion. The BaoBaoLe milk was a kind of artificial milk. As suggested, we have added the ingredients of it in the revised manuscript, namely, “These piglets did not take colostrum and were fed with the artificial milk (BaoBaoLe, Centree bio-tech (Wuhan) Co., LTD, Wuhan, China) including ≥20% crude protein, ≤0.5% crude fiber, 0.5%-1.1% calcium, ≥0.4% total phosphorus, ≤9% coarse ash, 0.4%-1.5% sodium chloride, and ≥1.6% lysine”. Please see the lines 81-85, page 2.
Q3: - What is TCID50? Explanation is required.
Response 3: Thanks so much for your kind suggestion. The TCID50 indicated 50% tissue culture infective dose, which was used for estimating the virus content. As suggested, we have added these in the revised manuscript, namely, “Out of them, 10 individuals were infected with 1 mL PEDV at three-days age, and the remaining five as the control group were handled by 1 mL of 0.9% normal saline (Bei-jing Epsilon Biotechnology Co., LTD, Beijing, China), in which, the actual PEDV titer of challenge dose was 10 50% tissue culture infective dose (TCID50)/mL. TCID50 was used for estimating the virus content.”. Please see the lines 85-89, page 2.
Q4: - Mention the dissection day (7) of the control group in the Materials and Methods, too.
Response 4: Thanks so much for your kind suggestion. As suggested, we have added the dissection day of the control group, namely, “Within the next four days, five piglets died at Day6 to Day7, and were classified as the susceptible group. The remaining five piglets were distributed to resistance group. For the piglets of control and resistance groups, we injected xylazine hydrochloride injection (0.2ml/Kg; Beijing Lab Anim Tech Develp Co., LTD, Beijing) into each individual for anaesthesia and dissection at Day 7.”. Please see the lines 94-98, pages 2-3.
Q5: - Citation for Primer 3 is needed.
Response 5: Thanks so much for your kind suggestion. As suggested, we have added the citation of Primer3, namely, “Primer 3 [20]”and “Untergasser A, Cutcutache I, Koressaar T, Ye J, Faircloth BC, Remm M, Rozen SG: Primer3--new capabilities and interfaces. Nucleic Acids Res 2012, 40(15):e115”. Please see the citation 20 and lines 111, page 3.
Q6: - Check if the authors want to use 4D-DIA or 4DIA. NF-kappaB or NF-κB and so on. There were many. Please make it consistent.
Response 6: Thanks so much for your kind suggestions. We are very sorry for the writing errors. As suggested, we have revised the “4DIA” to “4D-DIA”, and revised the “NF-kappaB” to “NF-κB”. Please see the lines 149 and 480, pages 4 and 15.
Q7: - Please rewrite the first sentence(?) in Lines 365-366.
Response 7: Thanks so much for your kind suggestion. As suggested, we have rewritten the sentence to “Some metabolites and proteins had significant differences in resistance vs susceptible and control vs susceptible comparisons, while had no significance in control vs resistance comparison.”. Please see the lines 392-394, page 11.

Reviewer 3 Report
Comments and Suggestions for Authors
The manuscript “Proteomic and metabolomic profiling elucidate the impact of PEDV on Yorkshire piglets and reveal the underlying molecular mechanism of PEDV response “ is an original research work in which authors showed the relathioship of some etabolites with a viral infection in piglets. I suppose it a valuable paper, interesting subject for audience, especially for researchers of genetic resistance to virus infections.
Throughout paper, I suggest replace “death” with “susceptible” group. Resistance and susceptible is more used is similar scientific context.
Material and methods: please present descriptive statistics of animals (15) live body weight, birth weight, BW of dams? parity of dams? sex of animals?
What was the possible contamination of mothers and fathers with this disease? Has there been resistance to the virus in the memory of the immune system of some Dams?
Address please in materials and methods, discussion and results: How likely is it that there is resistance to the virus in the memory of the immune system of the resistant group?
Had you any information about pedigree/relationship among animals? I mean, how many were full-sibs, or half sibs among 15 animals?
L81-86: please rewrite more clearly. It seems mixed sentences of two groups.
L 235-238: Please clarify how many image is counted per sample? How many Villi is counted per sample? We need a statistic to compare significant differences among three groups.
L 295: … within one group à within each group
L 514-516: This sentence is unclear. Please give more explanation here about disease of farm. This part has not sufficient relathonship with the rest of discussion and paragraph.
Author Response
Re: agriculture-2701930
Dear Reviewer 3,
Thank you very much for providing an opportunity for us to revise our paper for your journal. We also wish to thank you for the positive comments and constructive suggestions on the manuscript. We have made the essential revisions suggested as follows.
Comments and Suggestions for Authors
The manuscript “Proteomic and metabolomic profiling elucidate the impact of PEDV on Yorkshire piglets and reveal the underlying molecular mechanism of PEDV response “ is an original research work in which authors showed the relathioship of some etabolites with a viral infection in piglets. I suppose it a valuable paper, interesting subject for audience, especially for researchers of genetic resistance to virus infections.
Q1: Throughout paper, I suggest replace “death” with “susceptible” group. Resistance and susceptible is more used is similar scientific context.
Response 1: Thanks so much for your kind suggestion. As suggested, we have revised the “death” to “susceptible” throughout the paper. Please see the revised manuscript.
Q2: Material and methods: please present descriptive statistics of animals (15) live body weight, birth weight, BW of dams? parity of dams? sex of animals?
Response 2: Thanks so much for your kind suggestions. The 15 animals were from three litters involved in the second parity, of which, ten of them were doll pigs, and five were sow pigs. The birth weight and dead weight were determined. As suggested, we have added the detailed description for the animals as follows:
“A total of 15 Yorkshire piglets involved in three litters were randomly selected from the Changping pig breeding farm of the Institute of Animal Science, Chinese Academy of Agricultural Sciences (Beijing, China).”. Please see the lines 77-79, page 2.
“In addition, we weighted the piglets when they were birth and dead. There is no difference in their birth weight and dead weight, while their lose weight had the significant differences between any two groups (P < 0.05) (Figure 2G).”. Please see the lines 253-257, page 7.
Figure 2. PEDV copy number, intestinal lesions, and changes of weight. A The PEDV copy numbers in six small intestine tissues. B The diamine oxidase (DAO) content in serum. C-E The paraffin sections (10×10) of jejunal tissues. F The length of villi. G The weight of piglets. Lost weight indicated the change between birth and death weight.
Q3: What was the possible contamination of mothers and fathers with this disease? Has there been resistance to the virus in the memory of the immune system of some Dams?
Response 3: Thanks so much for your kind comments and suggestions. These 15 animals and their mothers and fathers were from well-managed and well-raised farm, which had the normal immune routine. However, we fed these 15 animals with artificial milk when they were born, and they did not take colostrum. They did not have the virus in the memory of the immune system.
Q4: Address please in materials and methods, discussion and results: How likely is it that there is resistance to the virus in the memory of the immune system of the resistant group?
Response 4: Thanks so much for your kind suggestions. As suggestions, we cleared these piglets did not take colostrum and did not have the virus in the memory of the immune system, namely, “These piglets did not take colostrum and were fed with the artificial milk (BaoBaoLe, Centree bio-tech (Wuhan) Co., LTD, Wuhan, China) including ≥20% crude protein, ≤0.5% crude fiber, 0.5%-1.1% calcium, ≥0.4% total phosphorus, ≤9% coarse ash, 0.4%-1.5% sodium chloride, and ≥1.6% lysine.”. Please see the line 81-85, page 2.
In addition, we have added the corresponding discussion in the revised manuscript as follows:
“The piglets used in this study did not take colostrum, and no virus was in the memory of the immune system. Therefore, phenotypic and mechanistic changes could reflect the process of PEDV infection.”. Please see the line 451-453, page 14.
Q5: Had you any information about pedigree/relationship among animals? I mean, how many were full-sibs, or half sibs among 15 animals?
Response 5: Thanks so much for your kind comments. These 15 animals were from three litters, namely, three groups of full-sibs. As suggested, we have added these in the revised manuscript, namely, “A total of 15 Yorkshire piglets involved in three litters were randomly selected from the Changping pig breeding farm of the Institute of Animal Science, Chinese Academy of Agricultural Sciences (Beijing, China)”. Please see the lines 77-78, page 2.
Q6: L81-86: please rewrite more clearly. It seems mixed sentences of two groups.
Response 6: Thanks so much for your kind suggestion. As suggested, we have rewritten this paragraph as follows:
“These piglets did not take colostrum and were fed with the artificial milk (BaoBaoLe, Centree bio-tech (Wuhan) Co., LTD, Wuhan, China) including ≥20% crude protein, ≤0.5% crude fiber, 0.5%-1.1% calcium, ≥0.4% total phosphorus, ≤9% coarse ash, 0.4%-1.5% sodium chloride, and ≥1.6% lysine. Out of them, 10 individuals were in-fected with 1 mL PEDV at three-days age, and the remaining five as the control group were handled by 1 mL of 0.9% normal saline (Beijing Epsilon Biotechnology Co., LTD, Beijing, China), in which, the actual PEDV titer of challenge dose was 10 50% tissue culture infective dose (TCID50)/mL. TCID50 was used for estimating the virus content. The PEDV (G2 mutant virus) was isolated from the jejunum of pigs in Beijing Liuma Pig Raising Technology Co., LTD in 2014 (Shunyi, Beijing, China), and has been used for the infected animal experiments [19].” Please see the lines 81-91, page 2.
Q7: L235-238: Please clarify how many image is counted per sample? How many Villi is counted per sample? We need a statistic to compare significant differences among three groups.
Response 7: Thanks so much for your kind comments and suggestions. For each individual, one image was conducted, and three villi were counted per sample. In addition, we did the statistical analysis, and found the degree of intestinal damage was susceptible group > resistance group > control group.
As suggested, we have added these in the result section, namely, “In addition, we performed the paraffin sections of jejunal tissue (Figure 2C-F), and found that the susceptible and resistance piglets had the obvious loss of intestinal villi than control piglets (P < 0.05). The susceptible showed the worst intestinal injury with lots of lost intestinal villi and congested interstitial vessels.”. Please see the lines 249-253, pages 6-7.
Figure 2. PEDV copy number, intestinal lesions, and changes of weight. A The PEDV copy numbers in six small intestine tissues. B The diamine oxidase (DAO) content in serum. C-E The paraffin sections (10×10) of jejunal tissues. F The length of villi. G The weight of piglets. Lost weight indicated the change between birth and death weight.
Q8: L295: … within one group à within each group
Response 8: Thanks so much for your kind suggestion. As suggested, we have revised “one” to “each”. Please see the line 319, page 9.
Q9: L514-516: This sentence is unclear. Please give more explanation here about disease of farm. This part has not sufficient relathonship with the rest of discussion and paragraph.
Response 9: Thanks so much for your kind suggestion. As suggested, we have deleted this sentence.

Round 2
Reviewer 2 Report
Comments and Suggestions for Authors
The authors have addressed most of my concerns.
Comments on the Quality of English LanguageOkay